# Dynamic fluctuations in a bacterial metabolic network

Shuangyu Bi[1,2], Manika Kargeti[1], Remy Colin [1], Niklas Farke[3], Hannes Link[3] & Victor Sourjik [1] ✉

The operation of the central metabolism is typically assumed to be deterministic, but dynamics and high connectivity of the metabolic network make it potentially prone to generating fluctuations. However, time-resolved measurements of metabolite levels in individual cells that are required to characterize such fluctuations remained a challenge, particularly in small bacterial cells. Here we use single-cell metabolite measurements based on Förster resonance energy transfer, combined with computer simulations, to explore the real-time dynamics of the metabolic network of *Escherichia coli*. We observe that steplike exposure of starved *E. coli* to glycolytic carbon sources elicits large periodic fluctuations in the intracellular concentration of pyruvate in individual cells. These fluctuations are consistent with predicted oscillatory dynamics of *E. coli* metabolic network, and they are primarily controlled by biochemical reactions around the pyruvate node. Our results further indicate that fluctuations in glycolysis propagate to other cellular processes, possibly leading to temporal heterogeneity of cellular states within a population.

Metabolism is central to cellular life, and metabolic and biosynthetic pathways are highly conserved among different species despite their enormous variety of lifestyles and ecological niches. Besides their immediate function in fueling all cellular activities, metabolism exerts an important regulatory control on many other cellular processes[1–3].

Metabolic networks contain multiple input-output interactions among transcription factors and other regulatory proteins, enzymes, metabolites, and fluxes[4]. Although central metabolism as a whole is thought to operate deterministically, cells are autocatalytic and stochastic systems in a dynamic equilibrium. Hence, the nature of metabolic networks and their interplay with gene expression and other cellular processes is inherently stochastic, which can lead to dynamic fluctuations in the network activity[5–12]. A well-studied source of cellular fluctuations is stochastic gene expression, which can generate substantial cell-to-cell variability in protein levels, either across an isogenic population or within one cell over time, and in extreme cases drives genetically identical cells into different physiological states[5,8,10,13–15].

Apart from fluctuations in gene expression, which occur on the timescales of minutes to hours, the ubiquitous post-translational regulation can lead to much faster fluctuations within protein networks. In contrast to stochasticity in gene expression, origins and physiological effects of such post-translational fluctuations remain little investigated[12]. One of the few studied examples of such stochasticity are activity fluctuations within bacterial chemotaxis pathway, which emerge from reaction stochasticity and extensive allosteric interactions among chemosensory proteins and might enhance environmental exploration[16,17]. Another example of a post-translational cellular process that is driven by dynamic instability within a network are oscillations of the central glycolytic pathway in eukaryotes. These glycolytic oscillations have been most intensively investigated, using NADH autofluorescence signal as a readout, in budding yeast *Saccharomyces cerevisiae* exposed to a combination of glucose and cyanide, primarily in synchronized cell populations and in cell lysates but also in single cells[18–22]. Glycolytic oscillations were observed in other eukaryotic cells, too[23–25]. It was proposed that this oscillatory process is

[1]Max Planck Institute for Terrestrial Microbiology and Center for Synthetic Microbiology (SYNMIKRO), D-35043 Marburg, Germany. [2]State Key Laboratory of Microbial Technology, Shandong University, Qingdao 266237, China. [3]University of Tübingen, D-72076 Tübingen, Germany. ✉ e-mail: victor.sourjik@synmikro.mpi-marburg.mpg.de

controlled primarily by allosteric regulation of phosphofructokinase, with substrate inhibition by ATP and product activation by AMP and fructose 1,6-bisphosphate[18,20,26–30], though additional interactions might be involved in regulation of such oscillations and their synchronization between individual cells[21,31–36]. Although glycolytic oscillations were also predicted to occur in prokaryotes[37–39], the only experimental indication for their existence in bacteria was up to now provided by early measurements of metabolite dynamics in the *Escherichia coli* bioreactor culture exposed to glucose[40].

In addition to these rapid fluctuations, recent studies have demonstrated slow metabolic fluctuations correlated with the cell cycle, both in eukaryotic and prokaryotic cells, possibly related to the cell-cycle variations in protein levels[41–43]. Moreover, metabolic diversification on the cell-cycle timescale can emerge within clonal bacterial populations due to stochastic, and sometimes regulated, variability in expression of metabolic genes and its effects on growth[7,44–49].

Fluctuations and resulting heterogeneity are commonly considered an impediment to the predictable performance of cellular networks, and most of these networks possess regulatory circuits that compensate undesirable effects of fluctuation on homeostasis[6,7,50]. However, in a number of cases cellular heterogeneity, including diversity of metabolic states, might also be beneficial for cell survival and growth, particularly in variable environments[14,51–54]. Studying metabolic variability might thus be important for our general understanding of metabolic regulation and the interplay among central cellular processes. But despite their potential significance, experimental quantification of single-cell metabolite levels with the necessary time resolution poses a number of challenges[9,11]. Nevertheless, the development of sensors based on fluorescence readouts, such as Förster (fluorescence) resonance energy transfer (FRET), has principally enabled such quantitative monitoring of metabolic processes in living cells[9,55–59].

In this work, we followed time-resolved concentration dynamics of the key metabolite pyruvate in single *E. coli* cells upon stimulation of the metabolic network, using a FRET-based sensor[60]. First, we show with a simplified kinetic model of metabolism that allosteric enzyme regulation in glycolysis can lead to sustained oscillations of pyruvate levels upon changes in the carbon source uptake, in a wide range of kinetic parameters. Consistent with these simulations, we experimentally demonstrate that the intracellular levels of pyruvate exhibit surprisingly large fluctuations on the timescale of ~100 s when starved *E. coli* cells are exposed to glucose or several other glycolytic carbon sources. We further observed that these metabolic fluctuations depend on the biochemical reactions involved in the conversion of pyruvate, and that they are likely coupled to the dynamics of other regulatory processes in a bacterial cell.

## Results

### A kinetic model of *E. coli* glycolysis predicts periodic oscillations

To first test the likelihood of oscillations in the levels of glycolysis metabolites in *E. coli*, as well as their timescale, we used a small kinetic model of *E. coli* glycolysis (Fig. 1a). The model consists of four metabolites and six metabolic reactions that were simulated with Michaelis-Menten and Hill kinetics (see Methods for a detailed description of the model). From a multitude of allosteric metabolite-enzyme interactions that regulate the activity of glycolysis enzymes, three of the most relevant ones were included in our model. The first interaction is the feedforward activation of pyruvate kinase (PYK) by fructose-1,6-bisphosphate (FBP), which plays an important role for glycolysis flux regulation in *E. coli*[61]. The other two interactions represent negative feedbacks from phosphoenolpyruvate (PEP) to the interconversion of hexose-phosphate and FBP, by respectively inhibiting phosphofructokinase (PFK) and activating fructose-1,6-bisphosphatase (FBPase), which together regulate the PFK-FBPase cycle[62].

As a starting point for the model analysis, we fixed the glycolytic flux to a constant value and randomly sampled all model parameters (maximal rates and binding constants) 10,000 times such that the model was at steady state. Next, we perturbed this steady state by decreasing the glucose uptake rate and analyzed whether this perturbation can lead to oscillations of the levels of pyruvate, the end product of our glycolysis model (Fig. 1b). A forward Fourier transformation of the time-dependent pyruvate levels revealed sustained oscillations of pyruvate concentrations for 440 of the tested 10,000 parameters sets. The typical period of oscillations across these simulations was several minutes, although some parameter sets caused faster or slower oscillations (Fig. 1c). Oscillations with a similar distribution of periods were also observed upon a strong increase in the glucose uptake starting from low steady state (Supplementary Fig. 1a–c), suggesting that predicted oscillatory dynamics are not specific to a particular type of perturbation.

To further identify network features that favor pyruvate oscillations, we first investigated the importance of individual allosteric feedbacks (Fig. 1d). The highest fraction of model parameters that yielded oscillations was observed for the model structure containing all three feedbacks, but simplified structures containing the positive allosteric regulation of PYK by FBP and either of the negative feedbacks from PEP to the PFK-FBPase cycle could also produce oscillations. We next compared the distribution of model parameters that yielded oscillations to all parameter sets for both downshift and upshift simulations (Fig. 1e, Supplementary Fig. 1d and Supplementary Data 1). Consistent with the structure analysis above, the oscillations were generally favored by the positive allosteric regulation of PYK by FBP (high *a3*). Low *Vmax* and low *Km* values of FBA, as well as several other model parameters were also promoting oscillations. Thus, in presence of a positive feedforward and at least one negative feedback, the stoichiometry and the kinetics of glycolysis can produce oscillations of intracellular metabolites on the timescale of several minutes in a broad range of parameter values and for both upshift and downshift of glycolysis flux.

### Measurements of intracellular pyruvate levels in *E. coli* cells using FRET sensor

We next established an assay to monitor dynamics of intracellular levels of pyruvate in stimulated *E. coli* MG1655 cells, using single-cell FRET microscopy in a flow-through chamber[16] in combination with the published FRET sensor[60] (Supplementary Fig. 2a and Methods). This FRET sensor employs a pyruvate-binding domain of *E. coli* pyruvate dehydrogenase repressor (PdhR) flanked by a cyan fluorescent protein (CFP) as a donor and a yellow fluorescent protein (YFP) as an acceptor for energy transfer (Fig. 2a). Pyruvate binding reduces energy transfer from the donor to the acceptor fluorophore, leading to the increased ratio of CFP to YFP fluorescence (FRET ratio) that is thus proportional to pyruvate-induced changes in FRET (see Methods). Population of *E. coli* cells expressing this sensor, which was grown in Luria Broth (LB) medium and subsequently pre-equilibrated in the M9 salts buffer without a carbon source, responded to the addition of pyruvate in a dose-dependent manner (Fig. 2b). Sensor response was also observed in cells that were permeabilized by toluene to enable the free exchange of metabolites[63] (Fig. 2c), confirming that the change in FRET is caused by pyruvate binding to the sensor. The half-maximal sensor stimulation in permeabilized cells was achieved at $EC_{50}$ of ~400 µM (Fig. 2d), which is close to the previously measured $Kd$ value of this sensor[60]. Much lower $EC_{50}$ (~6 µM) was observed for intact *E. coli* cells, likely due to an active uptake of pyruvate through the high-affinity BtsT symporter[64,65] that leads to a much higher cellular concentration of pyruvate compared to the exterior.

Changes in pyruvate levels were also observed within seconds upon a step-like addition and subsequent removal of glucose (Fig. 2e, Supplementary Fig. 2b, Supplementary Fig. 3). Such rapid conversion

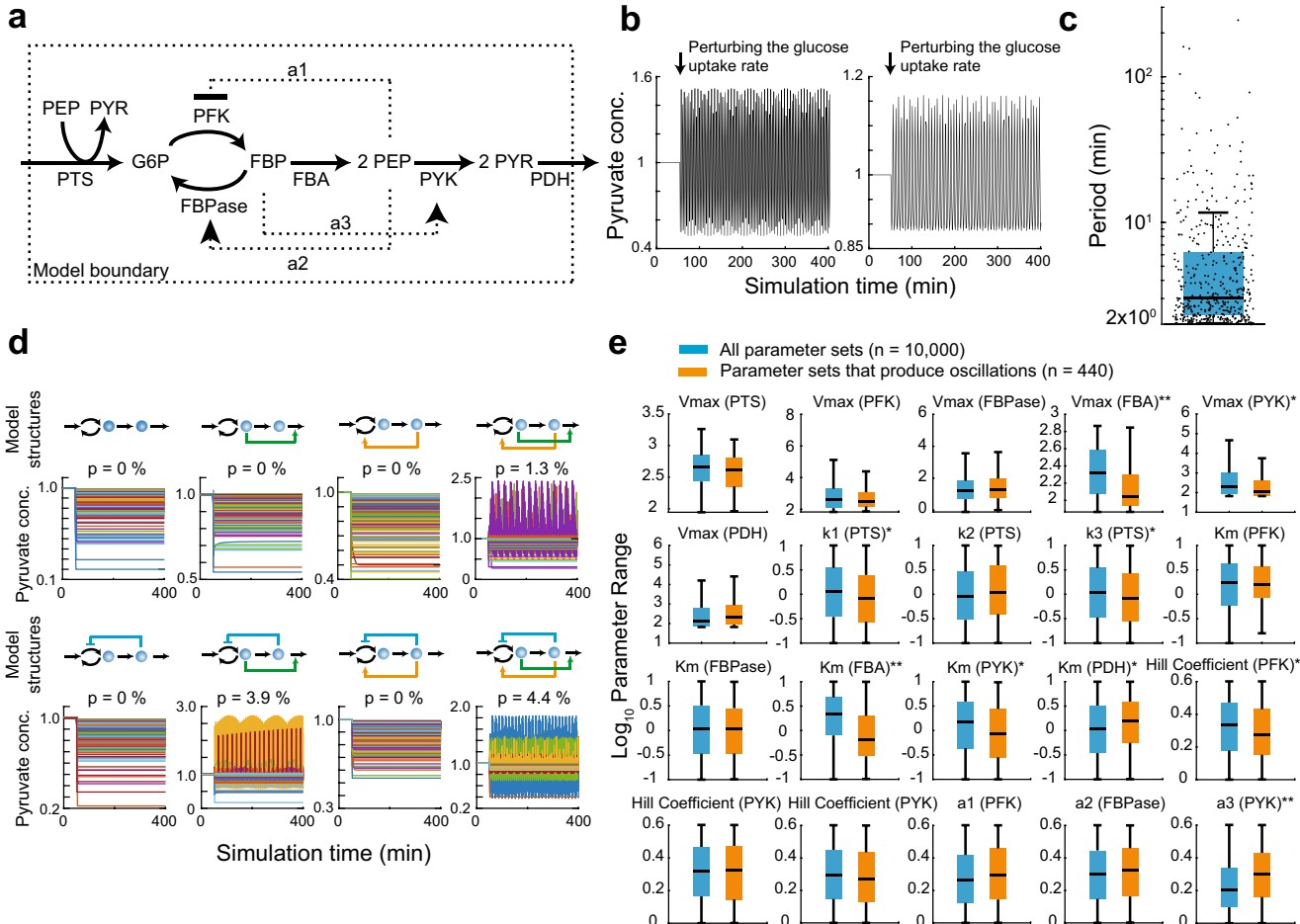

**Fig. 1 | A kinetic model of glycolysis predicts periodic oscillations. a Structure and stoichiometry of a simplified model of glycolysis.** The outer dotted line is the model boundary. Solid arrows are reactions and dotted arrows are allosteric interactions of metabolites with enzymes. G6P: glucose-6-phosphate, FBP: fructose-1,6-bisphosphate, PEP: phosphoenolpyruvate, PYR: pyruvate, PFK: phosphofructokinase, FBPase: fructose-1,6-bisphosphatase, PYK: pyruvate kinase, FBA: fructose-bisphosphate aldolase, PTS: phosphotransferase system, PDH: pyruvate dehydrogenase. **b** Two examples of simulated pyruvate concentrations with two different parameter sets that produce pyruvate oscillations. The model was initially at steady state and at t = 50 min the glucose uptake rate (Vmax of PTS) was decreased by 5%. **c** Boxplot showing the distribution of the periods of 440 simulations with oscillating pyruvate levels. Each black dot corresponds to a parameter set. The black solid line within the blue box denotes the median of the distribution. Boxes contain 50% and whiskers 99% of the simulated parameter values. **d** Ensemble modelling approach to test the influence of different feedback structures. Metabolites are indicted by blue spheres and metabolic reactions are indicated by black arrows. Feedback regulation is colored in green (FBP −> PYK), orange (PEP −> FBPase), and blue (PEP −| PFK). Feedback regulation was switched off by setting the corresponding power law exponent to zero. 10,000 stable steady states were sampled for each model and at t = 50 min the glucose uptake rate (*Vmax* of PTS) was decreased by 5%. P is the fraction of parameter sets that show oscillations. Only 1000 out of 10,000 parameter sets were plotted. **e** Orange boxes show parameter values in 440 sets that led to oscillations. Blue boxes show all 10,000 parameter sets that were random sampled. The black solid line within each box denotes the median of the distribution. Boxes contain 50% and whiskers 99% of the simulated parameter values. Asterisks denote parameters that moderately ($10^{-10} < p$-value $< 10^{-5}$, a = 0.01) and double asterisks denote parameters that strongly ($p$-value $< 10^{-10}$, a = 0.01) affect the model's propensity to produce pyruvate oscillations according to two-sided two-sample t-test. Exact $p$-values are listed in Supplementary Data 1.

of glucose to pyruvate agrees with the previously characterized response of glycolytic fluxes, which occurs on the timescale of seconds[66]. The $EC_{50}$ of glucose stimulation was ~5.5 µM (Fig. 2d), similar to the $EC_{50}$ value observed for the glucose uptake through the phosphotransferase system (PTS)[67]. The pyruvate sensor also responded to other glycolytic carbon sources fructose and glycerol, to a gluconeogenic carbon source acetate, as well as to several glycolytic intermediates, including glucose-6-phosphate (G6P), glyceraldehyde-3-phosphate (GAP), 3-phosphoglycerate (3PG), 2-phosphoglycerate (2PG) and phosphoenolpyruvate (PEP) (Fig. 2f, g, Supplementary Fig. 2c, Supplementary Fig. 3, Supplementary Fig. 4), which indicates rapid conversion of all these metabolites to pyruvate. Nevertheless, significantly higher (millimolar) concentrations of these metabolites, compared to glucose or pyruvate, were required to induce changes in pyruvate levels. Although the reason for this difference is not entirely

clear, lower sensitivity to these metabolites might be either due to their inefficient uptake, e.g. because of low expression or absence of specific uptake systems in *E. coli* grown in LB medium, or because of their lower entry points into glycolysis. The response dynamics and the magnitude of change in the FRET signal were also metabolite-specific. No response to any tested concentration of succinate was observed under our experimental conditions, suggesting that its metabolism does not lead to measurable changes in the levels of pyruvate, again possibly due to repression of the succinate transporter under our experimental conditions[4].

Notably, FRET responses to even saturating concentrations of glucose or other metabolites were smaller than the response to pyruvate itself (Fig. 2b, g, Supplementary Fig. 2c), suggesting that elicited increases in the levels of pyruvate always remain below the saturation of the sensor. The sensor was also appearently capable of detecting

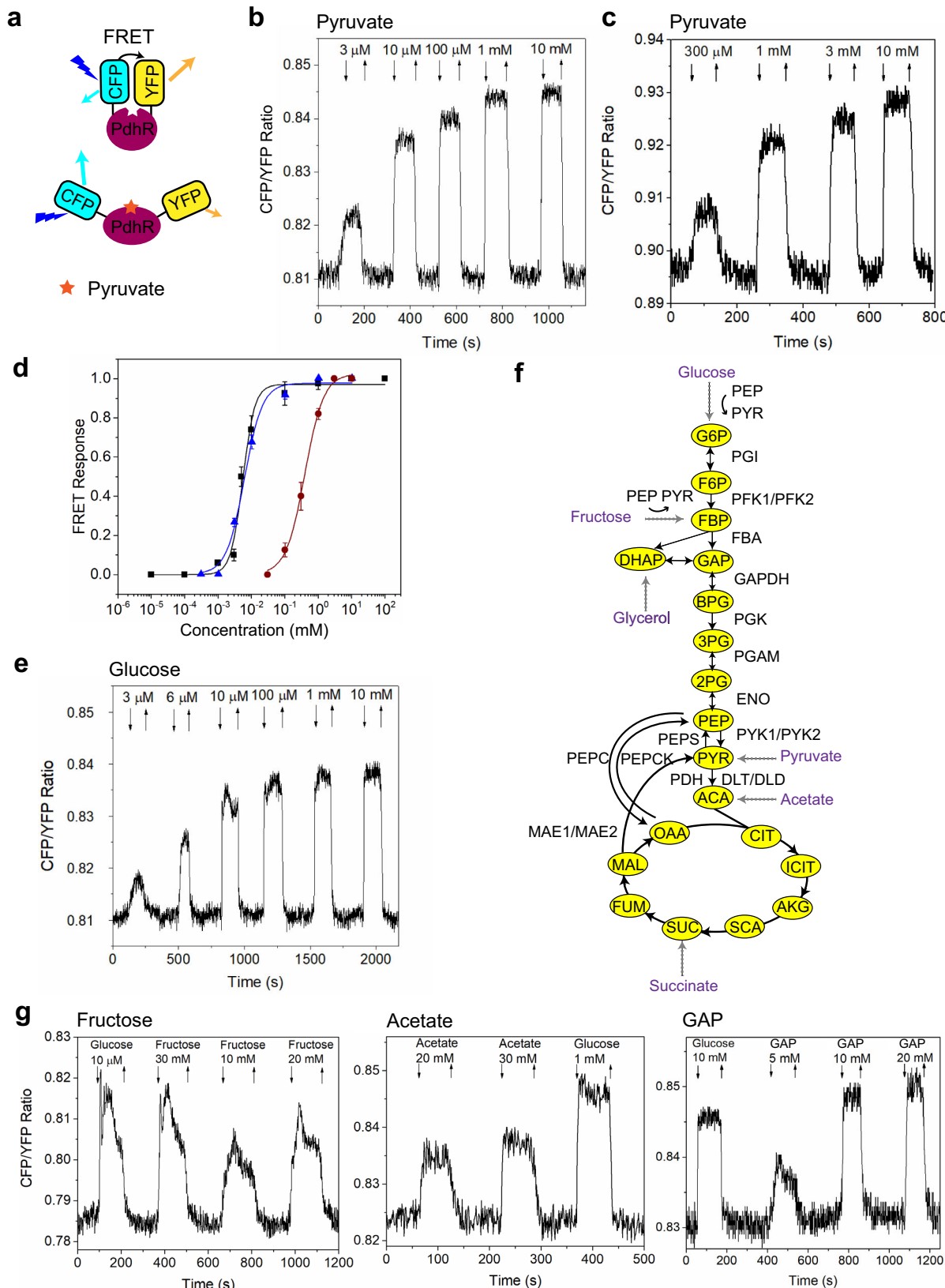

decrease in pyruvate levels below the baseline observed in the M9 buffer, since deenergizing cells by addition of 2,4-dinitrophenol (2,4-DNP) decreased the ratio of CFP to YFP fluorescence, with CFP signal decreasing and YFP signal increasing, as expected for a specific FRET response (Supplementary Fig. 2d). However, it has to be noted that only qualitative conclusion could be drawn from this experiment, since

exposure to 2,4-DNP also led to an additional unspecific increase of fluorescence in both YFP and CFP channels as seen in the negative controls that express either the FRET sensor with strongly reduced sensitivity to pyruvate[60] or a direct fusion between CFP and YFP (Supplementary Fig. 2e, Supplementary Fig. 2f and Supplementary Table 1). This unspecific increase, although different from the effect of

**Fig. 2 | In-vivo and in-vitro FRET measurements of pyruvate levels in *E. coli*.**
**a** Schematic representation of the PdhR-based pyruvate sensor used in this study. Pyruvate binding leads to increased separation between CFP and YFP fused to different termini of PdhR, which reduces resonance energy transfer from excited CFP to YFP and therefore increases the ratio of CFP to YFP fluorescence. **b** An example of FRET measurement for a population of intact *E. coli* cells expressing the pyruvate sensor. Cells were grown in LB medium, harvested and immobilized in a flow chamber under steady flow of M9 buffer (Supplementary Fig. 2a). After initial adaptation, cells were stimulated by addition and subsequent removal of indicated concentrations of pyruvate in M9 buffer, as shown by down and up arrows. **c** The FRET measurement of response to pyruvate for *E. coli* cells that were permeabilized by treatment with toluene. **d** The dose dependences of the pyruvate sensor response to pyruvate (blue triangles) or glucose (black squares) in intact cells, as well as of the response in permeabilized cells (red circles). Plotted are changes in the CFP/YFP ratio, normalized to the maximum change at saturating stimulation. Data are presented as means of three independent biological replicates ± SD. **e** The FRET measurement of response of intact cells to indicated concentrations of

glucose. **f** Schematic illustration of the *E. coli* central glycolysis pathway and the tricarboxylic acid (TCA) cycle. G6P: glucose-6-phosphate, F6P: fructose-6-phosphate, FBP: fructose-1,6-bisphosphate, GAP: glyceraldehyde-3-phosphate, DHAP: dihydroxyacetone phosphate, BPG: 1,3-bisphosphoglycerate, 3PG: 3-phosphoglycerate, 2PG: 2-phosphoglycerate, PEP: phosphoenolpyruvate, PYR: pyruvate, ACA: acetyl-CoA, CIT: citrate, ICIT: isocitrate, AKG: α-ketoglutarate, SCA: succinyl-CoA, SUC: succinate, FUM: fumerate, MAL: malate, OAA: oxaloacetate, PGI: phosphoglucose isomerase, PFK: phosphofructokinase, FBA: fructose-bisphosphate aldolase, GAPDH: glyceraldehyde phosphate dehydrogenase, PGK: phosphoglycerate kinase, PGAM: phosphoglycerate mutase, ENO: enolase, PYK: pyruvate kinase, PDH: pyruvate dehydrogenase, DLT: dihydrolipoamide transacetylase, DLD: dihydrolipoamide dehydrogenase, PEPS: PEP synthase, MAE: malic enzyme, PEPC: PEP carboxylase, PEPCK: PEP carboxykinase. Both glucose and fructose are transported by the phosphotransferase system (PTS), driven by conversion of PEP to PYR. **g** Examples of the FRET responses of intact cells to indicated concentrations of fructose, acetate, and GAP, with glucose used as control.

---

2,4-DNP on the functional pyruvate sensor, was not exactly proportional between channels and affected the ratio of CFP to YFP fluorescence, precluding exact measurement of the change in the level of pyruvate in this case.

## Dynamics of intracellular pyruvate levels in populations and in single cells

We next followed response dynamics of the pyruvate sensor upon prolonged step-like exposure of buffer-equilibrated (i.e. starved) *E. coli* cells to glucose. For subsaturating, micromolar concentrations of glucose, the observed initial changes in pyruvate levels were highly dynamic (Fig. 3a, b and Supplementary Fig. 5a). At a level of cell population, the response appeared as strongly damped oscillations, with an initial overshoot and subsequent undershoot relative to the average level of pyruvate that was reached upon prolonged stimulation. In contrast, stimulation with strongly saturating, millimolar levels of glucose elicited simple increase in the pyruvate levels without any apparent subsequent dynamics (Fig. 3c, d).

We further observed that both the initial response and subsequent dynamic changes in the levels of pyruvate could be distinguished in individual cells, although in this case the FRET ratio measurements were expectedly noisier than in the cell population (Fig. 3a–d and Supplementary Fig. 5a). Whereas the population-average pyruvate level rapidly stabilized after glucose addition, even at the intermediate (10–100 μM) levels of glucose where damped oscillations were observed, the majority of individual cells continued to exhibit large fluctuations in the FRET ratio for the entire duration of the observation, >1000 s after the initial stimulation. These single-cell fluctuations become apparently desynchronized and they were therefore averaged at the population level. Because these fluctuations were observed only at sub-millimolar levels of glucose but not in cells that were equilibrated in buffer or stimulated with millimolar levels of glucose (Fig. 3c, d and Supplementary Fig. 5b), we conclude that they reflect genuine dynamics of pyruvate levels rather than the noise of single-cell measurements.

In order to quantitatively characterize these glucose-induced dynamic fluctuations of intracellular pyruvate levels, we first determined the frequency content of the fluctuations of the FRET ratio. This was done by computing the power spectral density (PSD) of the single-cell FRET ratio, $s_R(\omega)$, which reports the contribution of FRET ratio oscillations at a given frequency $\omega/2\pi$ to the overall FRET ratio fluctuations[16]. Consistent with their comparatively stable FRET ratio (Supplementary Fig. 5b), cells that were equilibrated in M9 buffer or exposed to millimolar levels of glucose showed constant low level of fluctuations over the entire frequency range (Fig. 3e). This indicates that residual FRET ratio fluctuations are caused by the white shot noise of the microscopy measurements. In contrast, for *E. coli* cells exposed

to 10 or 100 μM concentrations of glucose, the PSD at low frequencies ($\omega/2\pi$ <0.02 Hz) was clearly above the background noise (Fig. 3e, Supplementary Fig. 5c, d), consistent with large fluctuations of the FRET ratio on the timescale of several minutes. Although individual cells exhibited different levels of such low-frequency fluctuations at these sub-millimolar levels of glucose, the distribution of the low-frequency PSD values remained monomodal but shifted to higher values compared to cells equilibrated in buffer or exposed to millimolar levels of glucose (Supplementary Fig. 6).

We next calculated the autocorrelation of fluctuations in the single-cell FRET ratio for *E. coli* cells exposed to 10 μM or 10 mM glucose, as well as for buffer-adapted cells (Fig. 3f). As expected for the uncorrelated shot noise, no pronounced autocorrelation beyond zero lag time was observed for buffer-adapted cells or in cells exposed to 10 mM glucose. In contrast, the FRET ratio for cells stimulated with 10 μM glucose had an autocorrelation that was consistent with damped oscillations, with an apparent periodicity of ~180 s. This periodic pattern became even more apparent when autocorrelation was calculated for the FRET signals measured using higher (100×) magnification objective (Supplementary Fig. 5d, e). Notably, in this case also an expected peak in the PSD value at ~0.006 Hz could be distinguished. Nevertheless, for subsequent experiments we continued to use the 40× magnification objective to enable larger field of view and thus higher statistics.

As an additional negative control for the specificity of the observed fluctuations, we utilized the aforementioned FRET sensor with strongly reduced sensitivity to pyruvate[60] (Supplementary Table 1). Indeed, *E. coli* cells carrying this sensor could not respond to 10 μM glucose (Supplementary Fig. 7a), and they exhibited only low level of fluctuations over the entire frequency range (Supplementary Fig. 7b). These results further confirm specificity of the FRET responses and of the single-cell pyruvate fluctuation observed upon stimulation with intermediate levels of glucose.

Since our model predicted oscillatory behaviour upon both upshift and downshift in glucose levels, we further tested the dynamics of intracellular pyruvate levels in response to the downshift from high (1 mM) to intermediate (100 μM) concentration of glucose. Our results confirmed that such downshift could also induce fluctuations (Supplementary Fig. 8a, b). However, these fluctuations were less pronounced than those observed upon a glucose upshift, indicating that starved *E. coli* cells might be more prone to exhibit the oscillatory dynamics of the metabolic network.

## Dependence of pyruvate fluctuations on supplied metabolites, genetic perturbations, and energy state of the cell

In order to better understand the nature of observed pyruvate fluctuations, we investigated whether they could be induced by exposure

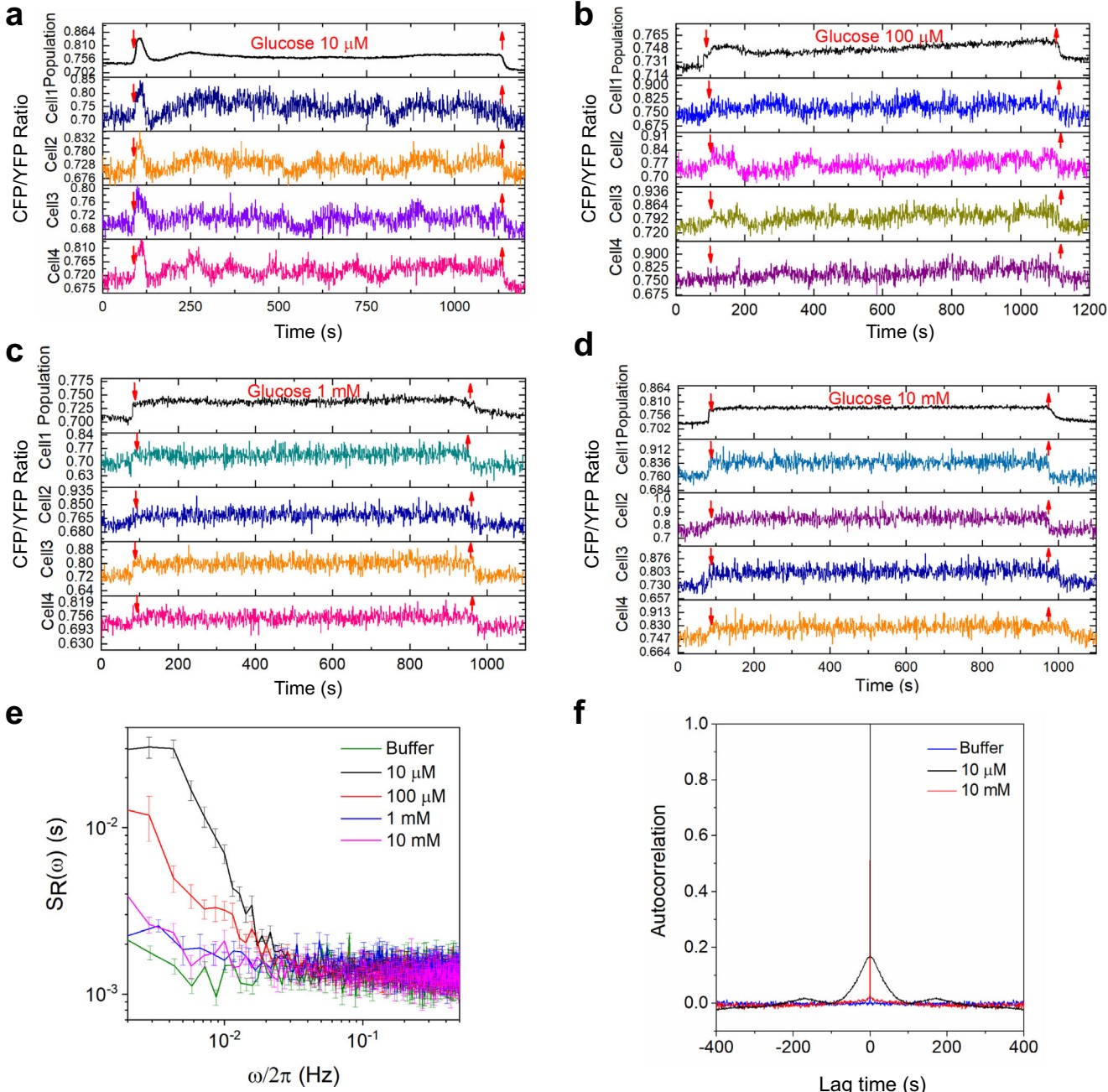

**Fig. 3 | Dynamics of intracellular pyruvate levels in individual *E. coli* cells exposed to glucose.** The population-averaged (black) and typical single-cell (colors) FRET measurements of pyruvate levels. Cells were first equilibrated in M9 buffer and subsequently stimulated by addition and removal of 10 μM (**a**), 100 μM (**b**), 1 mM (**c**), or 10 mM glucose (**d**) at indicated time points. The average PSD (**e**) and autocorrelation (**f**) of the FRET ratio fluctuations in individual cells adapted in buffer or exposed to indicated concentrations of glucose. The error bars in (**e**) represent standard errors of the mean (SEM). The sample sizes are 83 (10 μM), 139 (100 μM), 89 (1 mM), 91 (10 mM), and 75 (buffer) single cells, from three independent biological replicates in each case.

to carbon sources other than glucose or to glycolytic intermediates at concentrations that elicited subsaturating responses, smaller or comparable to the response towards 10 μM glucose (Fig. 2g and Supplementary Fig. 4). As discussed above, these concentrations were metabolite-specific and substantially higher than for glucose or pyruvate, possibly either due to their inefficient uptake or lower points of entry into glycolysis. When stimulated with fructose, which is transported by the fructose-specific branch of phosphotransferase system (PTS) (Fig. 2f), *E. coli* cells exhibited fluctuations in pyruvate levels that were only somewhat weaker than those induced by glucose (Fig. 4a). However, dynamics of the PTS system could not be the main source of the observed fluctuations, since they could be also induced by addition

of G6P, which is taken up independently of the PTS (Fig. 2f and Fig. 4a). Distributions of the low-frequency PSD values for these metabolites remained monomodal but shifted to higher values, as was already observed for glucose stimulation (Supplementary Fig. 6, Supplementary Fig. 9a–g), indicating that induced fluctuations are not limited to a subpopulation of cells. Exposure to lower glycolysis metabolite GAP induced weak fluctuations, and only residual low-frequency fluctuations could be observed upon addition of 2PG. Addition of glycolysis end-products PEP and pyruvate, as well as of acetate produced no fluctuations above the shot noise at tested concentration of these metabolites (5 mM–20 mM for PEP, 3 μM–1 mM for pyruvate, and 20 mM–30 mM for acetate). The pyruvate fluctuations were also

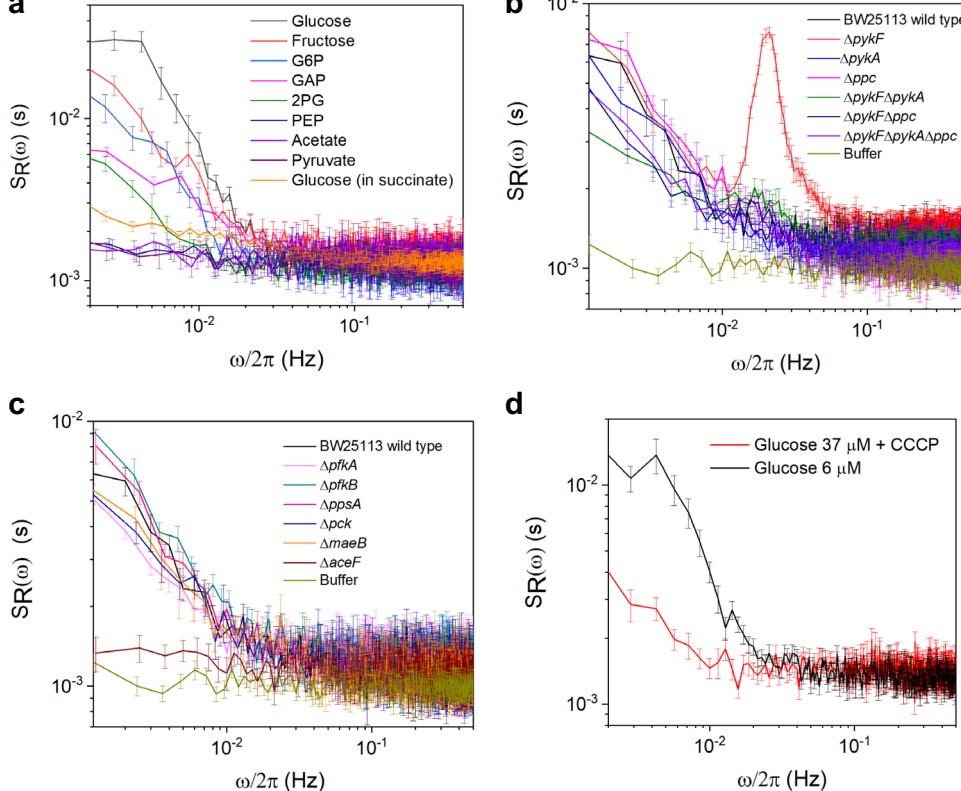

**Fig. 4 | Dependence of pyruvate fluctuations on carbon sources, genetic perturbations and energy state. a** The average PSD of the FRET ratio fluctuations for the cells in the presence of different carbon sources and metabolic intermediates. Used concentrations were 10 µM glucose, 30 mM fructose, 15 mM G6P, 7 mM GAP, 20 mM 2PG, 5 mM PEP, 30 mM acetate and 10 µM pyruvate. For testing the glucose response in presence of succinate, cells were first pre-equilibrated in 10 mM succinate and then stimulated by 10 mM glucose in presence of 10 mM succinate. The sample sizes are 75 (fructose), 66 (G6P), 105 (GAP), 55 (2PG), 125 (PEP), 69 (acetate), 75 (pyruvate), 83 (glucose) and 236 (glucose in succinate) single cells. **b, c** The average PSD for the FRET ratio fluctuations of *E. coli* BW25113 wild type and its indicated knockout strains upon exposure to glucose. The glucose concentration for Δ*aceF* was 1 mM, and for the other strains 10 µM. The sample sizes are 145 (wild type), 337 (Δ*pykF*), 120 (Δ*pykA*), 106 (Δ*ppc*), 369 (Δ*pykF*Δ*pykA*), 314 (Δ*ppc*Δ*pykF*), 343 (Δ*ppc*Δ*pykF*Δ*pykA*), 94 (Δ*pfkA*), 93 (Δ*pfkB*), 97 (Δ*aceF*), 124 (Δ*ppsA*), 71 (Δ*maeB*), 67 (Δ*pck*), and 179 (buffer) single cells. **d** The comparison of the average PSD of the pyruvate fluctuations with and without CCCP, upon exposure to indicated glucose levels that give a similar response (Supplementary Fig. 13b). The sample sizes are 94 (CCCP-adapted) and 172 (without CCCP). In **a–d**, the error bars represent standard errors of the mean (SEM).

largely suppressed when cells were exposed to glucose in presence of succinate that stimulates the TCA cycle without directly affecting the levels of pyruvate under our conditions (Fig. 4a, Supplementary Fig. 9h, Supplementary Fig. 10), indicating that cellular energy state might play a role in the observed behavior.

The observed differences between abilities of individual metabolites to induce fluctuations at least partly match the predictions of our computational model. Indeed, lower amplitude of fluctuations induced by GAP compared to G6P, and lack of fluctuations upon exposure to PEP, are consistent with the predicted importance of the PEP-dependent regulation of the PFK-FBPase cycle. Other possible contributions to oscillations, such as dynamics of the energy state of the cell, were not explicitly considered by our simple model and thus could not be compared (see Discussion).

We further systematically studied the dependence of fluctuations on the deletion of individual glycolytic enzymes, with particular focus on the metabolic reactions involved in production and consumption of pyruvate, using the KEIO collection of *E. coli* gene knockouts[68]. Upon exposure to glucose, the parental strain of the KEIO collection, BW25113, expressing the FRET sensor showed pyruvate fluctuations that were comparable to those in the MG1655 background (Fig. 4b and Supplementary Fig. 11a). Among the reactions involved in the conversion of PEP to pyruvate, the deletion of *pykF* encoding the pyruvate kinase 1 (PYK1), the primary pyruvate kinase in *E. coli*[69], markedly increased periodic oscillations of the FRET signal in individual cells

upon exposure to 10 µM glucose (Fig. 4b and Supplementary Fig. 11b). Although low-frequency fluctuations were similar in this background to those in the wildtype cells, we observed a highly pronounced peak at an intermediate frequency of ~0.02 Hz. Consistently, the FRET ratio for these glucose-stimulated Δ*pykF* cells showed clear autocorrelation, with larger amplitude but also higher frequency than in the wildtype cells (Supplementary Fig. 11c, Fig. 3d, and Supplementary Fig. 5e). Also in this case, the distribution of the PSD levels in Δ*pykF* cells showed a monomodal shift to higher values (Supplementary Fig. 11e, f). The level of pyruvate and its upregulation upon exposure to glucose in Δ*pykF* strain were similar to that of the wild type and apparently within the dynamic range of the FRET sensor (Supplementary Fig. 11d and Supplementary Fig. 2).

Although the pyruvate fluctuations were not markedly affected by the single deletion of *pykA* encoding pyruvate kinase 2 (PYK2) or *ppc* encoding PEP carboxylase (PEPC) (Fig. 4b, Supplementary Fig. 12a), both these knockouts, individually or combined, reduced fluctuations in the Δ*pykF* background below the wildtype level (Fig. 4b, Supplementary Fig. 11g–i, Supplementary Fig. 12a). Moreover, the pyruvate fluctuations were abolished by the deletion of *aceF* that encodes one of the components for the pyruvate dehydrogenase complex, pyruvate dehydrogenase (PDH), which converts pyruvate to acetyl-CoA (Fig. 4c, Supplementary Fig. 12b, c). Although deletions of these enzymes might affect the steady-state level of pyruvate, which was apparently the case for *aceF* knockout as judged from markedly higher levels of FRET in

buffer-equilibrated cells (Supplementary Fig. 12d), in all of tested strains the levels of pyruvate were still responsive to the addition of glucose and thus likely within the working range of the FRET sensor (Supplementary Fig. 12d, e). Nevertheless, higher concentration of glucose (1 mM) had to be used to induce changes in pyruvate levels in the Δ*aceF* mutant, and even this stimulation was still subsaturating (Supplementary Fig. 12d).

The fluctuations were not apparently affected by deletions of other tested glycolytic genes, including *pfkA* or *pfkB* (Fig. 4c and Supplementary Fig. 12b), indicating that – in contrast to the proposed mechanism of glycolytic oscillations in yeast[21] – regulation of the phosphofructokinase does not seem to play an important role for pyruvate oscillations in *E. coli*.

Finally, we studied how pyruvate fluctuations are affected when the energy state was lowered by treating *E. coli* cells with proton gradient uncoupler carbonylcyanide-m-chlorophenylhydrazone (CCCP). In contrast to the treatment with high concentration (1 mM) of 2,4-DNP (Supplementary Fig. 2d), the level of pyruvate in buffer was not apparently affected by exposure to 10 μM CCCP. Cells also remained responsive to glucose, albeit with decreased sensitivity (EC$_{50}$ ~18.5 μM). However, the amplitude and frequency of pyruvate fluctuations were largely reduced upon CCCP exposure (Fig. 4d), possibly related to the apparently slower response to glucose in presence of CCCP (Supplementary Fig. 13a, b).

### Fluctuations in activity of the PTS and in intracellular cAMP and NADH levels

The strongest pyruvate fluctuations in the wildtype cells were observed for glucose and fructose (Fig. 4a), which are both substrates of the PTS that couples sugar uptake to its phosphorylation through a series of phosphotransfer reactions that use PEP as a phosphodonor and generating pyruvate as a byproduct (Fig. 2f)[3]. Although stimulation with G6P excluded the PTS as the sole source of fluctuations (see above), the PTS might nevertheless play a role in the observed dynamic behavior, as proposed previously[40].

We thus directly investigated the dynamics of the PTS system upon exposure to glucose, using the same experimental setup as for the pyruvate sensor. The phosphorylation state of the PTS proteins is known to be reduced as a consequence of sugar transport. This affects associations between multiple PTS proteins, as well as inhibitory interactions of the glucose-specific PTS components EIIA$^{Glc}$ with several non-PTS transporters that could be measured by FRET[67]. Consistent with this previous work, when *E. coli* cells expressing a FRET pair of EIIA$^{Glc}$ fused to CFP and the galactose transporter MglA fused to YFP were grown in tryptone broth (TB), pre-equilibrated in M9 buffer and subsequently stimulated by glucose, we observed enhanced interaction between these proteins (Fig. 5a, b). These changes in FRET were also visible in individual cells. Similar to the pyruvate sensor, single-cell levels of FRET for the EIIA$^{Glc}$-CFP / MglA-YFP pair showed increased dynamics even after prolonged exposure to glucose, indicating sustained fluctuations in the PTS phosphotransfer reactions. The PSD and autocorrelation analyses revealed that these fluctuations had a similar frequency range compared to the pyruvate fluctuations (Fig. 5c, Fig. 3e, Supplementary Fig. 14a–c). Glucose-induced fluctuations were apparently specific, since fluctuations in FRET for this EIIA$^{Glc}$-CFP / MglA-YFP pair were much weaker in cells that were either equilibrated in M9 buffer or exposed to pyruvate, though pyruvate also elevated the FRET signal. Fluctuations at similar frequency were observed for cells grown in LB and subsequently tested in M9, although their amplitude was somewhat lower (Supplementary Fig. 14d). These results suggest that the fluctuations in pyruvate levels are likely to be at least partly coupled to fluctuations in the phosphorylation state of PTS proteins.

As phosphorylated EIIA$^{Glc}$ also activates adenylate cyclase to synthesize second messenger cyclic adenosine monophosphate (cAMP),

we next investigated whether the intracellular cAMP level could fluctuate along with the pyruvate level and the PTS activity in *E. coli* cells exposed to glucose. This was done using a FRET sensor based on *E. coli* cAMP receptor protein Crp that is flanked by CFP and YFP (Fig. 5d). Stimulation by glucose that lowers cAMP synthesis, and thus the fraction of cAMP-bound Crp, expectedly increased the ratio of CFP to YFP fluorescence in the population and in individual cells (Fig. 5d, e). Also for the cAMP sensor, single-cell levels of FRET showed increased fluctuations upon exposure to 10 μM glucose, and these fluctuations had a similar frequency range to the pyruvate and PTS activity fluctuations (Fig. 5f, Fig. 3e, Supplementary Fig. 15), indicating their possible connection. Consistently, cAMP fluctuations were much weaker in cells that were either adapted in M9 buffer or exposed to a high (10 mM) level of glucose.

NADH is generated during conversion of GAP to BPG, as well as in the downstream TCA cycle, and it is commonly used as a readout for glycolytic oscillations in yeast[21,70]. We thus analyzed the fluctuations of the intracellular NAD(P)H levels by measuring its autofluorescence in *E. coli* MG1655 cells, as done previously[41]. NAD(P)H levels indeed increased upon addition of glucose (Fig. 5g). We also observed fluctuations of the NAD(P)H autofluorescence for individual cells adapted in glucose, in a similar frequency range as the pyruvate fluctuations (Fig. 5h, i, Supplementary Fig. 16). Thus, the NAD(P)H fluctuations might also be connected to the pyruvate fluctuations.

## Discussion

Like many other protein reaction networks, metabolic networks are stochastic in nature and potentially prone to fluctuations[9,11]. As metabolism ultimately drives all cellular processes, fluctuations and instability of metabolic pathways could have a profound impact on cellular physiology. Several metabolic processes were indeed proposed to oscillate or fluctuate. The best-established example are glycolytic oscillations that can be observed in yeast and also in mammalian cells[20,23–25,71]. Glycolytic oscillations are typically induced in yeast by exposure of starved cells to glucose combined with cyanide, although weaker and less sustained oscillations could also be induced by glucose alone[32,71], and they primarily originate due to the allosteric regulation of PFK[18,27,28,30]. However, whether such glycolytic oscillations are common and physiologically significant remains unclear[26,33].

Here we adapted the FRET-based single-cell microscopy[16], which combines high sensitivity and second-scale temporal resolution necessary to characterize stochastic network fluctuations, to measure dynamics of the key metabolite pyruvate in *E. coli*. Our key observation is that the intracellular pyruvate levels in individual *E. coli* cells exhibit large fluctuations on the timescale of several minutes upon stimulation with intermediate (sub-millimolar) levels of glucose and other glycolytic carbon sources. In wildtype cells, these fluctuations occurred at frequencies below 0.02 Hz, showing limited periodicity of ~3 min. This timescale is much shorter than that of gene expression noise, and the fluctuations could be observed in non-growing cells, meaning that they must originate from dynamics of the metabolic network at the post-translational level. Their broad frequency range strongly indicates that these fluctuations reflect an interplay between multiple metabolic reactions and not dynamics of a single circuit, which is not surprising given the large number of regulatory metabolite-enzyme interactions[66,72,73].

Confirming the inherent potential of *E. coli* metabolic network to fluctuate, our computer simulations demonstrated that even a small simplified kinetic model of *E. coli* glycolysis could produce pyruvate oscillations upon changes in the glucose uptake, for a sizeable fraction of tested parameter values. Notably, while oscillation frequency varied dependent on the particular parameter set, the overall range as well as the average value of predicted frequencies were similar to the experimentally observed frequency spectrum of fluctuations. The analysis of the model structure revealed that a positive feedforward

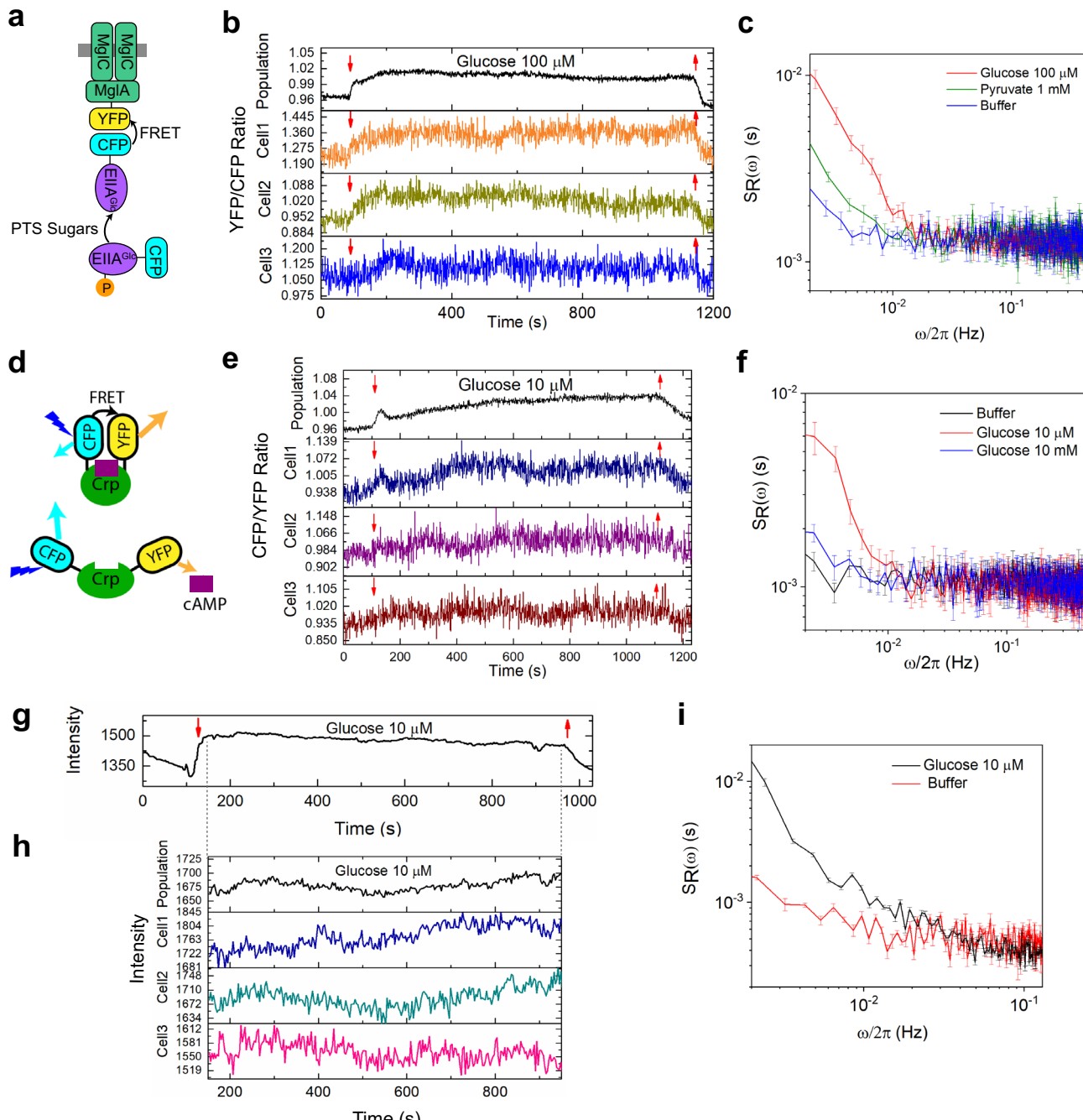

**Fig. 5 | Fluctuations of the PTS activity and the intracellular cAMP and NAD(P)H level in _E. coli_ cells exposed to glucose. a** Schematic representation of the EIIA$^{Glc}$-CFP/MglA-YFP FRET pair used in this study as the readout of the PTS activity. Glucose transport leads to enhanced interaction of EIIA$^{Glc}$ with MglA, which increases resonance energy transfer from excited CFP to YFP and therefore increases the ratio of YFP to CFP fluorescence. **b** Time course of population-averaged (black) and typical single-cell (colors) measurements of the FRET ratio for _E. coli_ MG1655 cells expressing the EIIA$^{Glc}$-CFP/MglA-YFP pair. Cells were first equilibrated in buffer and subsequently stimulated by addition and subsequent removal of 100 µM glucose, as indicated by down and up arrows. **c** Corresponding average PSD of the FRET ratio fluctuations for the EIIA$^{Glc}$-CFP/MglA-YFP pair in presence of glucose, pyruvate or buffer, as indicated. Cells were grown in tryptone broth (TB). The sample sizes are 170 (glucose, 100 µM), 93 (pyruvate) and 72 (buffer) single cells. **d** Schematic representation of the Crp-based cAMP sensor

used in this study. **e** Time course of population-averaged (black) and typical single-cell (colors) measurements of the FRET ratio for _E. coli_ MG1655 cells expressing the cAMP FRET sensor. **f** Corresponding average PSD of the FRET ratio fluctuations for the cAMP sensor in presence of different concentration of glucose or buffer, as indicated. The sample sizes are 60 (glucose, 10 µM), 83 (glucose, 10 mM), and 103 (buffer) single cells. **g** Time course of population-averaged intracellular NAD(P)H autofluorescence of MG1655 cells. Cells were first equilibrated in buffer and subsequently stimulated by 10 µM glucose. **h** Time course of 59 cells averaged (black) and typical single-cell (colors) measurements of intracellular NAD(P)H autofluorescence in MG1655 cells exposed to 10 µM glucose. Single-cell traces were tracked and analyzed from 150 s to 950 s corresponding to the time course of (**g**), as indicated by dash lines. **i** Corresponding average PSD of NAD(P)H autofluorescence fluctuations in presence of glucose or in buffer. The sample sizes are 59 (glucose) and 41 (buffer). In **c, f, i**, the error bars represent standard errors of the mean (SEM).

regulation of pyruvate kinase by FBP and a negative feedback regulation of G6P conversion to FBP by PEP are required to obtain oscillations upon downshift in the glucose uptake. Moreover, the kinetic parameters of the fructose-bisphosphate aldolase, that uses FBP as a substrate, were important determinants of oscillations. Such potential significance of the FBP-dependent regulation might be consistent with our experimental observation that the pyruvate fluctuations were smaller when cells were stimulated with GAP (product of FBA) compared to the stimulation with G6P or fructose that enter glycolysis upstream of FBP. Moreover, exposing *E. coli* to PEP, pyruvate or to the gluconeogenic carbon source acetate produced no fluctuations, in agreement with the role of PEP-dependent regulation and involvement of the pyruvate kinase and/or dehydrogenase.

The importance of the reactions around the pyruvate node, mediated by the pyruvate kinases 1 (PYK1) and 2 (PYK2) and pyruvate dehydrogenase (PDH), and possibly also by PEP carboxylase (PEPC), in inducing metablic fluctuations was also supported by our studies of genetic perturbations. Deletion of *aceF* that encodes one of the components of PDH was the only identified mutation that completely abolished pyruvate fluctuations in our experiments. We hypothesize that increased levels of pyruvate in this knockout might either suppress fluctuations by interfering with the regulatory feedbacks in glycolysis, or mask them due to high pyruvate background. In contrast, deletion of *pykF* led to largely enhanced and faster pyruvate oscillations compared to the wild type. This enhancement could be abolished by further deletion of either *pykA* or *ppc*, indicating that increase metabolic flow through one or both of these might cause stronger oscillations. Although increased oscillations in the knockout of *pykF*, which is known to be positively regulated by FBP[74], was unexpected, FBP was reported to positively regulate PEPC[75] that might create an alternative feedforward loop with a different frequency of oscillations. Nevertheless, even a strain deleted for all of *pykF*, *pykA* and *ppc* still exhibits pyruvate fluctuations upon stimulation with glucose, showing that other reactions must also contribute to this phenomenon.

One of such contributors might be the dynamics of PTS, which couples sugar uptake to the conversion of PEP to pyruvate. A previous study reporting oscillatory dynamics of several metabolites, including pyruvate, in a fermenter-growing culture of *E. coli* cells, hypothesized that such fluctuations might be caused by the PTS[40]. Indeed, the largest amplitude of fluctuations in our experiments was observed for glucose, followed by another PTS substrate fructose. Moreover, protein interactions within the PTS exhibited activity-dependent fluctuations on a similar timescale upon exposure to glucose. However, pronounced pyruvate fluctuations were also observed in response to G6P and also to GAP, which are not PTS substrates, clearly ruling out the PTS dynamics as the only or primary source of fluctuations.

Apparent coupling of the PTS dynamics to pyruvate fluctuations might be nevertheless physiologically important, given multiple regulatory functions of the PTS in bacteria[3,67,76]. Indeed, intracellular level of the second messenger cAMP, which depends on the PTS activity, show fluctuations similar to those of the PTS system. Since the cAMP-dependent transcription factor Crp regulates the expression of a large number of genes that are involved in carbon catabolism and growth control, its fluctuating activity might have a major impact on transcriptional control in *E. coli*.

Finally, although our simulations could yield periodic pyruvate oscillations even without taking the energy or redox state of the cell into account, cellular energy level seems to be important for the experimentally observed dynamics. The pyruvate fluctuations upon exposure to glucose were largely suppressed in presence of succinate and, in contrast to the model predicitons, the magnitude of induced fluctuations was higher for nutrient upshift than for the downshift in glucose levels, suggesting that well-energized cells are less prone to fluctuations than starved cells. On the other hand, the fluctuations markedly slowed down upon exposure to the uncoupler CCCP. At least

in the latter case, this might be related to much slower increase in the pyruvate level upon cell exposure to glucose in presence of CCCP, which could effectively filter out high-frequency fluctuations. Indeed, the rate of sugar uptake was shown to affect glycolytic oscillations in yeast[35]. Fluctuations in the levels of intracellular NAD(P)H were also observed under our experimental conditions, which occurred in a similar frequency range and are thus likely related to the pyruvate fluctuations.

Concluding, using the single-cell FRET microscopy enabled us to observe large minute-scale fluctuations in the output of the glycolysis in *E. coli* upon exposure to glucose and other carbon sources. Despite their similarity to the glycolytic oscillations in yeast and other eukaryotes, which are typically monitored using NADH autofluorescence[20] but can also be observed at the level of pyruvate[24], the determinants of the pyruvate fluctuations observed in *E. coli* are different from the established eukaryotic models. While the regulation of PFK is believed to be the main contributor to the glycolytic oscillations in yeast, the fluctuations in *E. coli* appear to have multiple origins within glycolysis, with biochemical reactions involved in the production or consumption of pyruvate, which are subject to multiple allosteric feedback mechanisms[66], being of central importance. Given the multitude of regulatory functions of glycolytic intermediates, and of NAD(P)H levels, the PTS activity and of cAMP levels to which the observed pyruvate fluctuations seem to be coupled, the observed dynamic might have profound effects on bacterial metabolism, gene regulation and cell physiology that remain to be explored.

## Methods

### The kinetic model of *E. coli* glycolysis
The stoichiometry of our model is shown in Fig. 1a. Mass balancing results in a system of ordinary differential equations (ODEs), F, which is a temporal function of the state variables $x$ (G6P, FBP, PEP, PYR) and the kinetic parameters $p$. In total, the system comprises 4 variables and 21 kinetic parameters. Dilution of metabolites by growth was not considered due to large differences between growth dilution and glycolytic flux.

$$F(x,p) = \frac{dx}{dt} = \begin{cases} PTS + FBPase - PFK \\ PFK - FBPase - FBA \\ 2 \cdot FBA - PYK - PTS \\ PTS + PYK - PDH \end{cases} \quad (1)$$

The six reactions (PTS, PFK, FBPase, FBA, PYK, PDH) are described by the following kinetic equations:

Reaction1 describes the PTS-mediated uptake of glucose from outside the system boundary depending on the ratio of PYR/PEP. With a glucose uptake rate of 8 mmol g$^{-1}$ h$^{-1}$ and a specific cell volume for *E. coli* (2 µl mg$^{-1}$) the reaction rate for the PTS system is:

$$r1 = PTS = \frac{Vmax1}{k1 \cdot \frac{PYR}{PEP} + k2 + k3 \cdot \frac{PYR}{PEP} + 1} = \frac{8\,mmol\,g^{-1}h^{-1}}{0.002\,l\,g^{-1}} \cdot \frac{1h}{60\,min} = 66.66\,mmol\,l^{-1}min^{-1}$$

(2)

Reaction 2 (PFK) follows Hill-type kinetics as it was shown that the enzyme exhibits cooperative kinetics towards its substrate (n1 is the Hill coefficient). The enzyme is allosterically inhibited by PEP which is modelled by a negative power-law term with the exponent.

$$r2 = PFK = \frac{Vmax2}{\left(1 + \frac{Km1}{G6P}\right)^{n1}} \cdot PEP^{-a1} \quad (3)$$

Reaction 3 (FBPase) is modelled by Michaelis-Menten type kinetics. The activation of FBPase by PEP is modelled by a positive

power-law term:

$$r3 = \text{FBPase} = Vmax3 \cdot \frac{\text{FBP}}{\text{FBP} + Km2} \cdot \text{PEP}^{a2} \qquad (4)$$

The flux ratio between PFK and FBP is randomly sampled between 0.01 and 1 and constraint to a net flux of 66.66 mmol l$^{-1}$ min$^{-1}$.

Reaction 4 (FBA) is modelled by Michaelis Menten type kinetics. Here, glycolysis is simplified by condensing four reactions that convert FBP into PEP into a single reaction:

$$r4 = \text{FBA} = Vmax4 \cdot \frac{\text{FBP}}{\text{FBP} + Km3} \qquad (5)$$

Reaction 5 (PYK) follows Hill-type kinetics. The allosteric feed-forward activation by FBP is modelled by a power law:

$$r5 = \text{PYK} = \frac{Vmax5}{\left(1 + \frac{Km4}{\text{PEP}}\right)^{n2}} \cdot \text{FBP}^{a3} \qquad (6)$$

Reaction 6 (PDH) follows Hill-type kinetics:

$$r6 = \text{PDH} = \frac{Vmax6}{\left(1 + \frac{Km5}{\text{PYR}}\right)^{n3}} \qquad (7)$$

All state variables (i.e. metabolites) were set to 1. The system was initialized to a steady state by first setting the total reaction flux of all net reactions to the glucose uptake rate. Then, parameter values of binding constants ($k1$-$k3$, $Km1$-$Km5$), Hill coefficients ($n1$–$n3$) and power-law exponents (exponents $a1$-$a3$) were inserted. Each power-law corresponds to an allosteric feedback loop, and we created eight models to account all possible feedback structures. To remove a feedback loop, the respective power-law exponent was set to 0. Then the maximum velocities were calculated ($Vmax1$-$Vmax6$). Binding constants were sampled between 0.1 and 10, Hill coefficients were sampled between 1 and 4, where a coefficient of 1 resembles a Michaelis-Menten type kinetic. Power-law exponents for allosteric interactions were sampled between 1 and 4. To assess the stability of the steady state, we tested whether the eigenvalues of the Jacobian matrix are negative ($\lambda < -10^{-5}$) and therefore stable. If the steady state was unstable, the parameter set was discarded, and a new parameter set was sampled from a log-uniform distribution. This procedure was repeated until 10,000 stable steady states were achieved.

The perturbation of the glucose uptake was simulated by changing the uptake rate by 5% at 50 min simulation time. The resulting time-course data were then processed to identify parameter sets which led to oscillations. First, a polynomial first order fit was performed to remove trends and align the time courses. Second, the data were Fourier transformed from time domain to frequency domain. Signals with amplitudes above 0.001 and the corresponding parameter sets were then selected.

## Cell growth and sample preparation

Strains and plasmids used in this work are listed in Supplementary Table 1. The gene fragment of pyruvate FRET sensor was amplified from the plasmid pT162M104[60] and cloned using *Xba*I and *Sal*I restriction enzymes into the vector pTrc99A, to generate the plasmid pSB27. The non-responsive FRET sensor as a control was generated from pSB27, with mutations of V188D, V198M, and S245T. For the cAMP FRET sensor, the sequence of YFP-Crp-CFP was ligated to the *Nco*I and *Sal*I digested pTrc99A to generate pVS1503. *E. coli* strains carrying the plasmid pSB27 encoding pyruvate FRET sensor or pVS1503 encoding cAMP sensor were grown at 30 °C overnight in Luria Broth (LB) supplemented with 100 µg ml$^{-1}$ ampicillin. The culture was subsequently diluted 1:100 in LB containing 100 µg ml$^{-1}$ ampicillin and

200 µM isopropyl-β-D-thiogalactopyranoside (IPTG) for induction of the sensor, and grown at 30 °C under vigorous shaking (200 rpm) until the OD$_{600}$ reached 0.6 to obtain the day culture. For incubation of *E. coli* MG1655 expressing EIIA$^{\text{Glc}}$-CFP/MglA-YFP FRET pair[67], Tryptone Broth (TB) or LB containing 100 µg ml$^{-1}$ ampicillin, 34 µg ml$^{-1}$ chloramphenicol, 200 µM IPTG and 0.1% arabinose were used to grow cells to OD$_{600}$ = 0.6 at 200 rpm, 30 °C. The day culture was harvested by centrifugation, washed twice with the M9 buffer (40 mM Na$_2$HPO$_4$, 22 mM KH$_2$PO$_4$, 8.5 mM NaCl, 18 mM NH$_4$Cl, 1 mM MgSO$_4$, 0.1 mM CaCl$_2$, pH 7.0) and stored at the room temperature before the measurements.

For preparation of permeabilized cells, *E. coli* cells expressing the pyruvate sensor were resuspended in the PdhR assay buffer (10 mM Tris-HCl, 150 mM NaCl, pH 7.8). Permeabilization reagent toluene was added to the final concentration of 5 % ($v/v$) and then stirred gently on a rotary shaker (180 rpm) at the room temperature for 30 min. The pretreated cells were recentrifuged, washed twice with the PdhR assay buffer, and analyzed for the PdhR activity.

## Chemical compounds used in this study

D-fructose (Product No. F0127), glyceraldehyde-3-phosphate (Product No. G5251), 3-phosphoglycerate (Product No. P8877), 2-phosphoglycerate (Product No. 79480), phosphoenolpyruvate (Product No. P7002), acetate (Product No. S2889) and pyruvate (Product No. P2256) are from Sigma-Aldrich. Glucose-6-phosphate (Product No. SC-221489A) is from Santa Cruz Biotechnology. Glycerol (Product No. 3783.2) is from Carl Roth. All chemicals were tested for their purity to exclude contamination by glucose.

## Microscopy measurements

The FRET measurements were performed on an automated inverted microscope (Nikon Ti Eclipse, Nikon Instruments) controlled by the NIS-Elements AR software (version 4.40, Nikon Instruments)[16]. Briefly, *E. coli* cells expressing the pyruvate sensor, EIIA$^{\text{Glc}}$-CFP/ MglA-YFP FRET pair, or cAMP FRET sensor were attached to poly-lysine coated slides and were subsequently fixed at the bottom of a flow-through chamber. A constant flow (0.5 ml min$^{-1}$) of fresh M9 buffer and indicated concentrations of glucose, different carbon sources/intermediates or 2,4-DNP was used to stimulate cells. The cells were observed at 40× or 100× oil immersion objective lens and illuminated using a LED light (X-cite exacte, Lumen Dynamics). Images were continuously recorded with a 1.0 s integration time and 1.0 s exposure time in two spectral channels corresponding to CFP (472/30 nm) and YFP (542/27 nm) fluorescence using an optosplit (OptoSplit II, CAIRN Research) and the Andor Ixon 897-X3 EM-CCD camera (Andor Technology). For each single-cell FRET measurement, the field of view was chosen to contain both a small region of high-density cells and well-separated single cells. For each population FRET measurement, a region of high density with confluent cells was selected. Nikon perfect focus system was used during the measurements to maintain the focus. Experiments were performed at the room temperature (22 °C).

For measurements of the activity of pyruvate sensor in presence of CCCP, MG1655 cells expressing the sensor were pre-adapted in 10 µM CCCP and stimulated with different concentrations of glucose. For the intracellular NAD(P)H autofluorescence measurements, similar procedure was performed except that the DAPI filter (excitation 355/40 nm, emission 460/50 nm, beamsplitter 409 LP) and wildtype *E. coli* MG1655 without fluorescence were used to record the time course of the autofluorescent signals upon addition of glucose.

## Image processing and data analysis

The image analysis was performed using the NIS-Elements AR software (version 4.40, Nikon Instruments)[16]. Briefly, the CFP and YFP images were aligned with each other and a gray average of the two channels was delineated to create binary masks. For single-cell FRET

measurements, individual cells were detected by segmentation of the threshold image into individual objects, which resulted in a collection of distinct regions of interest (ROIs) for each frame of the movie. The NIS build-in tracking algorithm was then used to track the ROIs from frame to frame. The selected ROIs were then inspected manually and those not representing individual single cells or not well attached to the cover glass were discarded. Alternatively, in particular with movies with low signal over noise ratios, a Gaussian Blur was applied to the gray average of CFP and YFP channels, and the resulting movie was analyzed using the machine learning based image analysis software Ilastik v1.4.0rc2[77] to identify and track single cells and populations of cells. Pixels in each image were first classified between background and cells using the Pixel classification module. The resulting image was then separated in single cells and aggregated populations using the Object classification module. The resulting single-cell masks were then exported into ImageJ Fiji 2.0.0 where they were applied to the original registered CFP and YFP fluorescence channels. We then used a custom particle tracking software[78] to track individual cells and measure their average CFP and YFP fluorescence over time. Both methods of determination of mean CFP and YFP fluorescence traces in single cells gave similar results.

The average CFP and YFP values over the ROI corresponding to each tracked cell and the confluent population of cells were extracted as a function of time using MATLAB 8.4 R2014b (The MathWorks). The FRET ratio was computed as the ratio between CFP and YFP fluorescence for both the single cells and the population. For small chages in FRET as those observed here, this FRET ratio ($R$) is related to the negative change in FRET due to pyruvate binding ($\Delta F$) as $R = R_O - b\Delta F$, where $R_O$ is the FRET ratio for this sensor in absence of pyruvate and $b$ is a constant. Cells with a FRET ratio change of less than 10% of the population response were discarded as unresponsive. The percentage of such unresponsive cells was below 15% for any of our measurements. For the population FRET measurements, the FRET ratio was analyzed and plotted using KaleidaGraph v4.5 (Synergy Software). To obtain dose-response curves, data were fit to a Hill model $Y = A \times L^H/(L^H + K^H)$, where $Y$ is the FRET response (change in the FRET ratio), $L$ is the concentration of the stimulant, $A$ is the amplitude (for normalized response curves, $A = 1$), $H$ is the Hill coefficient, and $K$ is the $EC_{50}$. The data analysis for the NADH autofluorescent signals was similar instead of the gray average of only one channel was delineated and its value over the ROI was extracted.

### The PSD and autocorrelation analysis

The PSD and the autocorrelation analysis were carried out using MATLAB 8.4 R2014b. The camera dark count (106) was subtracted from each CFP and YFP time series. Each fluorescence time series was then corrected for bleaching, by fitting the part of the time series where cells are stimulated with an exponential decay ($f(t) = a\exp(-bt) + c$), or a linear one ($f(t) = a - bt$) in case of low bleaching, and dividing the time series by the fitted function. The FRET ratio was then computed as the ratio of corrected CFP divided by corrected YFP time series for each cell. For responsive cells, a $N = 800$ frames long subset of the ratio traces, starting about 50 s after the stimulation by the tested compound, was selected for further analysis of FRET ratio fluctuations. We note $r_{k,j}$, $k \in [1, N]$, this time series of ratio values for particle $j$ and $t_k = kdt$ the associated time stamps. Its discrete Fourier transform is defined as $\tilde{r}_{q,j} = \sum_k^N r_{k,j}\exp(-2\pi i \frac{qk}{N})$, with $q \in [1, N]$. Similar to our previous work, we compute the PSD as:

$$s_R\left(\omega = \frac{2\pi q}{N}\right) = \left\langle \frac{|\tilde{r}_{q,j}|^2}{(\bar{r}_{k,j})^2} \right\rangle_j \frac{dt}{N} \tag{8}$$

The unbiased autocorrelation function was computed using the build-in Matlab function as:

$$C(m) = \frac{1}{N - |m|} \sum_{k=0}^{N-|m|-1} \left\langle \frac{\delta r_{k+|m|,j}\delta r_{k,j}}{|\delta r_{k,j}|^2} \right\rangle_j \tag{9}$$

With $\delta r_{k,j} = r_{k,j} - \overline{r_{k,j}}$. In both cases, $\langle \cdot \rangle_j$ is an average over single cells and $\bar{x}_k = 1/N \sum x_k$ is a time average.

### Reporting summary

Further information on research design is available in the Nature Portfolio Reporting Summary linked to this article.

## Data availability

All the data generated in this study are provided within the paper, in the Supplementary Information and as Source Data files. Source data are provided with this paper.

## Code availability

The codes used for simulations in this study is available under [https://github.com/nfarke/Pyruvate_Oscillations]; [https://doi.org/10.5281/zenodo.7779240]. The code used for particle tracking is available under [https://github.com/croelmiyn/ParticleTracking]; [https://doi.org/10.5281/zenodo.7781880].

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

## Acknowledgements

We thank Alexandra Hahn for technical help and Tobias Erb for insightful discussions. We further thank Nicola Zamboni (ETH Zurich, Switzerland) and Silke Neumann for kindly providing the pyruvate and cAMP FRET sensors, respectively. This study was supported by the Max Planck Society (to V.S.) and the European Research Council (grant agreement no. 715650, ERC Starting Grant MapMe to H.L.).

## Author contributions

V.S. and S.B. conceived and designed the study; S.B., M.K., and R.C. conducted the experiments and analyzed the data; N.F., H.L. performed the computational modeling and simulations; all authors wrote the paper.

## Funding

## Competing interests

The authors declare no competing interests.
