## [Peer Review File · Nature Communications]

REVIEWER COMMENTS

Reviewer #1 (Remarks to the Author):

Summary

In their manuscript, Bi et al. show that in starved *E. coli*, a stepwise increase in glucose concentrations induce periodic fluctuations in the levels of pyruvate. These oscillations have a period of several minutes and are distinct from fluctuations resulting from stochastic gene expression. The authors show that a relatively simple kinetic model of *E. coli* glycolysis with only three allosteric metabolite-enzyme interactions (one feedforward and two negative feedback regulations) can generate such oscillations in response to a strong increase or decrease in glucose uptake rate on a scale that is comparable to what they found experimentally. The model indicates several features that favor the emergence of oscillations, such as a low K_m and low V_{max} for pyruvate dehydrogenase (PDH) and fructose-bisphosphate aldolase (FBA) as well as the positive allosteric regulation of pyruvate kinase (PYK) by fructose-1,6-bisphosphate (FBP).

With single-cell experiments using a FRET sensor for pyruvate, the authors show that the strength of the oscillations depends on glucose concentrations in the media, with the strongest signal at intermediate glucose levels (10-100 μM) and diminished oscillations at higher glucose levels (1-10 mM). The authors experimentally investigated where the pyruvate oscillations originate from by subjecting *E. coli* cells to different glycolytic intermediates and other carbon sources as well as by looking at deletion mutants of several genes encoding glycolytic enzymes. Notably, they found that multiple glycolytic reactions are contributing to the oscillations. Further experiments with a second FRET sensor revealed that already the phosphotransfer reactions in the phosphotransferase system (PTS) fluctuate upon exposure to glucose, suggesting that the PTS is also contributing to the pyruvate oscillations. Pyruvate oscillations were strongly reduced in the presence of succinate as well as upon lowering the energy state of the cells with a proton gradient uncoupler.

Lastly, the authors looked at NADH levels upon a glucose stimulus and found that these are also oscillating with a period similar to that of pyruvate, indicating that the oscillations are likely connected.

General comments

This manuscript describes a fascinating phenomenon that has previously been described in yeast and mammalian cells but has not yet received much attention in bacteria. The study presents a thorough and well-controlled investigation of glycolytic oscillations in *E. coli* with many novel insights. The conclusions and claims seem well supported by the data. While it remains open whether the metabolite

oscillations have a physiological impact or serve a role in metabolism and cell physiology, the presented findings establish important groundwork for future studies to build on.

The manuscript is well structured and clearly written. The experiments are clearly described, and the results are presented in useful figures and discussed appropriately.

Specific comments and suggestions

1. The model predicts that a downshift of high to low glucose levels also causes oscillations. Have the authors tested this experimentally as well? If so, what are the results?
2. Would it be possible to experimentally test the features predicted by the model to lead to emergence of oscillations (feedforward loop FBP-PYK, low K_m and low V_{max} of PDH and FBA) in an effort to pin down the molecular mechanism further?
3. Figure 1c/d: For the glucose downshift in silico experiments, the authors randomly sampled all model parameters 2000 times. However, in the glucose upshift experiments, 10,000 parameter sets were sampled? Would the findings for the glucose downshift stay identical with another 2000 randomly sampled parameter sets?
4. Line 232: For clarity, please specify here what is meant by intermediate glucose levels. (Personally, I would move Supplementary Figures 5b and c to main Figure 3, so that the traces across all 4 glucose concentrations can be easily compared - up to the authors.)
5. I would suggest that the authors extend the discussion to speculate on potential physiological consequences of these oscillations. E.g.: What are possible physiological consequences of such oscillations upon glucose stimuli? Could there be any advantages or disadvantages for the cells having these oscillations? Given that the oscillations affect not just pyruvate but also NADH (and possibly many more metabolite) levels, is it possible that the entire metabolic network in these cells is affected by the oscillations?

Reviewer #2 (Remarks to the Author):

In this manuscript Bi et al., first used kinetic modeling to anticipate periodic fluctuations in pyruvate levels and then used single-cell FRET microscopy to observe minute-scale fluctuations of pyruvate levels in *E. coli* upon exposure to glucose and other carbon sources.

Although I find the topic interesting, I have several major concerns with this work.

Overall, I fail to find an actual conclusion, besides the anecdotal observation of fluctuations in pyruvate levels (for which I have also major technical concerns, see later). What are or could be the functional implications of ~3minute scale fluctuations? Despite the authors claiming otherwise, there is a relative large body of direct experimental evidence demonstrating stochastic fluctuations in metabolism, and their relation to extrinsic noise in gene expression or their potential functional implications in bacterial physiology and the response to unexpected environmental challenges. What could be the functional role of such rapid fluctuations in pyruvate? How these fluctuations propagate through the network? Do they give raise to fluctuations also in flux through the TCA cycle?

It is also not clear why the authors focused exclusively on pyruvate. Are model simulations predicting larger and more robust oscillations for pyruvate with respect than other metabolites in the network?

What I find even more confusing is the attempt to explain pyruvate oscillations by affecting upstream or downstream metabolic processes in the cells, in what it seems a random process. It is not clear how much of the experimental evidence are consistent with or guided by model predictions, and hence whether model predictions are meaningful at all.

Finally, the observation of periodic oscillation in pyruvate using FRET sensor are not convincing. First it is not clear what is the signal-to-noise ratio in equilibrated cells and how significant are deviations from the mean observed in single cells. Most importantly the authors should measure oscillations also for other metabolites, ideally those predicted from the model to not oscillate. This will bring confidence on the measurements and strengthen the main observation.

Reviewer #3 (Remarks to the Author):

Summary

The work presents a detailed computational and experimental analysis of (structured) metabolic oscillations in *E. coli*, using cutting-edge single-cell-based tools. Overall, the manuscript is well written, and the results clearly presented. While the work undoubtedly contributes to the field of (microbial) metabolism, it is largely a descriptive study, adding to the growing body of knowledge that the metabolic behaviors of single cells should be viewed as stochastic and probabilistic processes. What the study does not address (and what the field currently needs) are mechanistic and (evolutionary) functional interpretations of phenomena like metabolic oscillations. The novelty in this work, therefore, lies mainly in a description of dynamic metabolic behaviors in single bacterial cells.

I recommend publication if the comments below are all satisfactorily addressed.

Noteworthy results

E. coli cells exhibit glycolytic oscillations under specific growth conditions, like those previously described for the yeast *S. cerevisiae*.

The use of FRET-based metabolite sensors provides high temporal resolution of metabolic dynamics in single bacterial cells.

Pyruvate levels fluctuate in single isogenic cells grown in the same environment, highlighting the occurrence of non-genetic metabolic heterogeneity in bacteria.

NAD(P)H fluctuations show fluctuations of a similar frequency to pyruvate, suggesting that these oscillations are related.

In contrast to *S. cerevisiae*, metabolic oscillations appear not to involve PFK, but rather enzymes involved in Pyruvate metabolism (with a mechanistic interpretation, that unfortunately, is quite vague).

Major comments

- The core model is used to demonstrate the capacity of (*E. coli*) glycolysis to generate metabolic oscillations and to identify parameters that affect the model's propensity to produce these oscillations. However, we do not learn all that much about the mechanistic underpinnings and especially, how the regulatory (allosteric) architecture affects this behavior. Extending their current analysis to consider (generally) the sufficiency or necessity of specific feedbacks and the underlying architecture to generate the observed oscillations, would be informative (the structure of the network, incl allosteric interactions, may be more important than the choice of kinetic parameters).

-Some FRET-sensor controls appear to be lacking (and should be included for publication):

o Control experiments with a non-responsive FRET-sensor are missing (e.g. pT162M115). This is needed to show that no other cellular parameters affect the sensor's response (Line 172, 184, 238).

o 2,4-DNP can significantly influence fluorescence ratios if not properly corrected for. How was the background contribution of 2,4-DNP accounted for? Here also, a non-responsive sensor would be needed.

o Did the authors calculate fluorescence bleed-through in their system, and was this corrected for? For reliable FRET-based measurements, this should be accounted for.

- Could the authors motivate their choice to sample from a (log)uniform distribution? Would a bell-shaped distribution (e.g. normal, log-normal) not be a better reflection of non-genetic metabolic heterogeneity? Given the large sampling range, are the parameters still biologically relevant, and if not could the authors motivate why this doesn't matter?

- Given how small the oscillation-inducing glucose range is ($< 100 \mu\text{M}$). How were concentrations for other carbon sources (Fig 4) chosen? It seems like these results may depend strongly on the exact concentration of the substrate. Without some variation in substrate concentration, the conclusion that some carbon sources induce oscillations and others don't is shaky. All that can be said, is that at the chosen concentrations these substrates don't induce oscillations (and they might at other concentrations). For example, if glucose experiments were done only at concentrations above 1 mM, the conclusion would be that glucose does not induce oscillations.

Minor comments

- Showing population averaged only (e.g. fig S7C does not show whether or not single cells display fluctuation behaviours, I also miss here the autocorrelation and PSD plots). Furthermore, showing only 4 cells is prone to cherry-picking. Perhaps an appendix with all the single-cell traces (at least for the most important figures, such as S7C, 3, 6C) or, more practically, a table indicating fractions of oscillating and non-oscillation single cells (also see next comment).

- Under conditions where oscillations are observed, it would be informative to indicate what fraction of single cell traces show oscillations and what fraction doesn't. I.e. is this a robust behaviour at the single cell level?

- The authors mention that bleaching correction was applied for autocorrelation and PSD analyses. What about the other experiments? 1.0 s exposure seems long, and with repeated measurements, this can certainly lead to significant bleaching.

- Line 233: what is a majority? Please give a % or fraction

- Line 378: can also be aspecifically induced by glucose? The use of a non-phosphorylatable eIIA_{glc} unit would be informative. Or measure the phosphorylation status in single cells? Or knockout and use another transport system?

- Line 380-381: For experiments with the PTS FRET pair, the authors use TB medium for “consistency” with previous experiments. From line 379, M9-medium grown cells display much reduced fluctuations, which would imply that something in LB/TB is important to induce large fluctuations – could the authors comment on what this may be, and whether this has implications for the interpretation?

- Line 391: do the authors see NADH fluctuations in the knockouts in S6g and S6h? And vice versa (the mutant in S6c?).

- Line 424: However, if such should read: However, whether such. In this sentence, “if” indicates a conditional, which is absent.

- Line 633: Cells with a FRET-ratio change of < 10% were considered unresponsive and discarded from analyses. Please provide an indication of the percentage/fraction of cells that were typically excluded. Does the inclusion/exclusion of these cells influence data interpretation? A small FRET-ratio does not necessarily indicate an unresponsive sensor, it could also just be that these cells do not experience large perturbations in pyruvate concentration.

- The link between the model and experiment could perhaps be made more explicit in the discussion. (it is only briefly discussed in the discussion, line 467). What is the model adding to the experiments, how well do experimental manipulations (e.g. knockouts) match model predictions?

- Could the authors comment on the observation that only PDH inactivation abolishes fluctuations? This step appears to exert a lot of control over the pathway behavior – what features are responsible for the dominant contribution of PDH (especially considering the conclusion that control appears to be distributed across several components).

- It is sometimes unclear why the authors choose specific and varying substrate concentrations in different experiments. E.g. Fig 4 and Fig S7 aceF mutant. 1 mM glucose is used, as opposed to 10 μM (in most other experiments). Especially, given the conclusion in Line 316-318 that deletion of aceF abolishes fluctuations (when exposed to 1 mM glucose), where experiments with WT showed that fluctuations are

absent at millimolar (incl 1mM) concentrations. In these instances, it would be useful to state clearly the reasons for these differences in substrate concentration.

- Could the authors comment/speculate on the functional significance of differences in the oscillatory behavior seen for 10 μ M vs 1 and 10 mM glucose pulses (as shown in Fig 3)? (also see next comment)

- It is e.g. striking that 30 mM of fructose induces oscillations, whereas 1mM of glucose doesn't. What (kinetic/regulatory) differences in the underlying metabolic network explain this stark difference in sensitivity?

- In the abstract, the authors conclude that fluctuations in glycolysis "apparently propagate to other cellular processes, leading to ... heterogeneity of cellular states within the population". The use of apparent implies observations that would support this statement. The data included here allows at best for speculation that fluctuations "could possibly propagate". Not all fluctuations have a functional impact on phenotypic states.

Point-by-point response to reviewers' comments

We would like to thank the reviewers for their thorough and critical review of the manuscript and for their overall positive appreciation of our work, as well as for their critical comments and suggestions for improvement of the manuscript. We have revised the manuscript according to the reviewers' suggestions. Below are our detailed responses to individual reviewers' comments:

Reviewer #1 (Remarks to the Author):

Summary

In their manuscript, Bi *et al.* show that in starved *E. coli*, a stepwise increase in glucose concentrations induce periodic fluctuations in the levels of pyruvate. These oscillations have a period of several minutes and are distinct from fluctuations resulting from stochastic gene expression. The authors show that a relatively simple kinetic model of *E. coli* glycolysis with only three allosteric metabolite-enzyme interactions (one feedforward and two negative feedback regulations) can generate such oscillations in response to a strong increase or decrease in glucose uptake rate on a scale that is comparable to what they found experimentally. The model indicates several features that favor the emergence of oscillations, such as a low K_m and low V_{max} for pyruvate dehydrogenase (PDH) and fructose-bisphosphate aldolase (FBA) as well as the positive allosteric regulation of pyruvate kinase (PYK) by fructose-1,6-bisphosphate (FBP).

With single-cell experiments using a FRET sensor for pyruvate, the authors show that the strength of the oscillations depends on glucose concentrations in the media, with the strongest signal at intermediate glucose levels (10-100 μM) and diminished oscillations at higher glucose levels (1-10 mM). The authors experimentally investigated where the pyruvate oscillations originate from by subjecting *E. coli* cells to different glycolytic intermediates and other carbon sources as well as by looking at deletion mutants of several genes encoding glycolytic enzymes. Notably, they found that multiple glycolytic reactions are contributing to the oscillations. Further experiments with a second FRET sensor revealed that already the phosphotransfer reactions in the phosphotransferase system (PTS) fluctuate upon exposure to glucose, suggesting that the PTS is also contributing to the pyruvate oscillations. Pyruvate oscillations were strongly reduced in the presence of succinate as well as upon lowering the energy state of the cells with a proton gradient uncoupler.

Lastly, the authors looked at NADH levels upon a glucose stimulus and found that these are also oscillating with a period similar to that of pyruvate, indicating that the

oscillations are likely connected.

General comments

This manuscript describes a fascinating phenomenon that has previously been described in yeast and mammalian cells but has not yet received much attention in bacteria. The study presents a thorough and well-controlled investigation of glycolytic oscillations in *E. coli* with many novel insights. The conclusions and claims seem well supported by the data. While it remains open whether the metabolite oscillations have a physiological impact or serve a role in metabolism and cell physiology, the presented findings establish important groundwork for future studies to build on.

The manuscript is well structured and clearly written. The experiments are clearly described, and the results are presented in useful figures and discussed appropriately.

We thank the Reviewer for this excellent summary of our findings and for the positive comments on our work.

Specific comments and suggestions

1. The model predicts that a downshift of high to low glucose levels also causes oscillations. Have the authors tested this experimentally as well? If so, what are the results?

We thank the Reviewer for this suggestion. We now performed the analysis of pyruvate dynamics in response to the downshift of glucose concentrations. The results are included as new Supplementary Figure 8. As discussed in the revised text (lines 304-310), there is indeed an induction of fluctuations by the downshift as predicted by the model, although these fluctuations are less pronounced than those induced upon glucose upshift. We hypothesize that this difference might be due to the effects of the energy state on pyruvate dynamics, which was not explicitly considered in our simple model of glycolysis.

2. Would it be possible to experimentally test the features predicted by the model to lead to emergence of oscillations (feedforward loop FBP-PYK, low K_m and low V_{max} of PDH and FBA) in an effort to pin down the molecular mechanism further?

As also suggested by the other referees, we have now performed additional analysis of the model structure to better pin down the mechanisms that are responsible for the emergence of oscillations in the model itself (new Figure 1d), and we revised the Discussion section (and also partly the Result section) to focus stronger on the comparison between the modeling and experimental results (lines 347-353, and lines 529-546). This comparison suggests that the model captures some but not all

determinants of the oscillatory behavior, which is not surprising given its simplicity. Direct testing the predictions of the model, e.g. by removing allosteric regulation or modifying enzymatic parameters, would not be simple given the complexity of metabolic regulation, and it will be an aim for our follow-up work.

3. Figure 1c/d: For the glucose downshift in silico experiments, the authors randomly sampled all model parameters 2000 times. However, in the glucose upshift experiments, 10,000 parameter sets were sampled? Would the findings for the glucose downshift stay identical with another 2000 randomly sampled parameter sets?

We thank the Reviewer for this observation. We now always use 10,000 parameter sets and we amended the parameter analysis with an analysis of the model structure in Figure 1.

4. Line 232: For clarity, please specify here what is meant by intermediate glucose levels. (Personally, I would move Supplementary Figures 5b and c to main Figure 3, so that the traces across all 4 glucose concentrations can be easily compared - up to the authors.)

We specified in the text that the intermediated glucose levels are 10-100 μM , and wherever relevant use “sub-millimolar” in addition or instead of “intermediate”. The former Supplementary Figures 5b and c were moved to Figure 3, as suggested.

5. I would suggest that the authors extend the discussion to speculate on potential physiological consequences of these oscillations. E.g.: What are possible physiological consequences of such oscillations upon glucose stimuli? Could there be any advantages or disadvantages for the cells having these oscillations? Given that the oscillations affect not just pyruvate but also NADH (and possibly many more metabolite) levels, is it possible that the entire metabolic network in these cells is affected by the oscillations?

We thank the Reviewer for pointing it out, and we agree that it is a highly interesting and important question. We have discussed it now more extensively in the Discussion section of the revised manuscript (lines 572-578 and 603-607). We indeed assume that at least a substantial fraction of the metabolic network and related regulatory interactions are affected by these oscillations. To further explore their extent, we performed additional experiments that show similar dynamic fluctuations of cAMP levels (new Figure 5d-f, Supplementary Figure 15, lines 451-463), which are likely caused by the dynamics of the metabolic and PTS networks. Since the cAMP-dependent transcription factor Crp regulates the expression of a large number of carbon-uptake and other genes in *E. coli*, such fluctuations might affect dynamics of gene activation on the timescale of minutes. Exploring potential benefits or negative effects of this dynamics for cell physiology will be the subject of our future research.

Reviewer #2 (Remarks to the Author):

In this manuscript Bi *et al.*, first used kinetic modeling to anticipate periodic fluctuations in pyruvate levels and then used single-cell FRET microscopy to observe minute-scale fluctuations of pyruvate levels in *E. coli* upon exposure to glucose and other carbon sources.

Although I find the topic interesting, I have several major concerns with this work.

Overall, I fail to find an actual conclusion, besides the anecdotal observation of fluctuations in pyruvate levels (for which I have also major technical concerns, see later). What are or could be the functional implications of ~3-minute scale fluctuations? Despite the authors claiming otherwise, there is a relatively large body of direct experimental evidence demonstrating stochastic fluctuations in metabolism, and their relation to extrinsic noise in gene expression or their potential functional implications in bacterial physiology and the response to unexpected environmental challenges. What could be the functional role of such rapid fluctuations in pyruvate? How these fluctuations propagate through the network? Do they give rise to fluctuations also in flux through the TCA cycle?

We thank the Reviewer for finding the topic interesting. As stated by the Reviewer, there is indeed a substantial body of literature on fluctuations in metabolism related to stochastic gene expression. We would like to emphasize that this has been already stated in the previous version of our manuscript, and the key studies were already cited (e.g. refs 16, 19 in the previous version of the manuscript). We have now expanded this discussion and cited additional literatures in the Introduction section (Ref. 45-49, 54).

In contrast to these previously observed gene-expression related fluctuations, in this manuscript we focus on the expression-independent fluctuations that are observed even in non-growing *E. coli* cells. As for this Reviewer's (and also other referees') questions about the extent and the functional implications of observed metabolic fluctuations, our previous experimental data already suggested that these fluctuations affect NAD(P)H levels and activity of the PTS network, which both have a number of regulatory functions in the cell. We have now further extended this experimental analysis, showing that similar dynamic fluctuations are also observed for the cAMP levels (new Figure 5d-f, Supplementary Figure 15, lines 451-463). Thus, the observed fluctuations likely propagate not only through the metabolic network but might have more general impact on cell physiology, particularly upon rapid changes in the nutrient supply. More generally, we absolutely agree that learning more about this impact and physiological significance would be exciting, and this constitutes the topic of our future research.

It is also not clear why the authors focused exclusively on pyruvate. Are model simulations predicting larger and more robust oscillations for pyruvate with respect than other metabolites in the network?

The reason to primarily focus on pyruvate was because it is the end product of the glycolysis pathway. We mention it now in lines 112-113. Other intermediate metabolites also oscillate in our model. And our experimental data, including additional experiments for cAMP added in the revision phase, show that fluctuations could also be observed at the levels of the PTS activity, NAD(P)H and cAMP.

What I find even more confusing is the attempt to explain pyruvate oscillations by affecting upstream or downstream metabolic processes in the cells, in what it seems a random process. It is not clear how much of the experimental evidence are consistent with or guided by model predictions, and hence whether model predictions are meaningful at all.

We admit that the correspondence between experiments and the model has not been discussed sufficiently in the previous version of the manuscript between. This has now been done in the Discussion section of the revised manuscript (lines 529-546). However, the choice of the metabolites and gene knockouts to be experimentally tested was systematic, covering different steps of glycolysis and more closely the reactions around the production and consumption of pyruvate. We apologize if this was not made entirely clear, and we explained it better in the text.

Finally, the observation of periodic oscillation in pyruvate using FRET sensor are not convincing. First it is not clear what is the signal-to-noise ratio in equilibrated cells and how significant are deviations from the mean observed in single cells. Most importantly the authors should measure oscillations also for other metabolites, ideally those predicted from the model to not oscillate. This will bring confidence on the measurements and strengthen the main observation.

Maybe it was not explained in sufficient clarity in the previous version of our manuscript, but because the low-frequency pyruvate fluctuations were not observed in cells that were equilibrated in buffer or stimulated with millimolar levels of glucose, we conclude that they reflect genuine dynamics of pyruvate levels rather than the noise of single-cell measurements. We stated it more clearly in lines 263-267 of the revised manuscript. Furthermore, we now performed an additional negative control experiment suggested by the Reviewer #3, showing that a mutated FRET sensor with strongly decreased sensitivity to pyruvate exhibits no response and no activity fluctuations when stimulated by intermediate level of glucose, confirming specificity of the observed response and fluctuations (Supplementary Figure 7). Although measurements of single-cell dynamics of other metabolite levels would be certainly highly interesting, we are currently limited by the availability of FRET sensors (we did try available FRET sensors for citrate and α -KG, but those have not shown sufficiently strong responses to be applicable for single-cell FRET measurements).

Reviewer #3 (Remarks to the Author):

Summary

The work presents a detailed computational and experimental analysis of (structured) metabolic oscillations in *E. coli*, using cutting-edge single-cell-based tools. Overall, the manuscript is well written, and the results clearly presented. While the work undoubtedly contributes to the field of (microbial) metabolism, it is largely a descriptive study, adding to the growing body of knowledge that the metabolic behaviors of single cells should be viewed as stochastic and probabilistic processes. What the study does not address (and what the field currently needs) are mechanistic and (evolutionary) functional interpretations of phenomena like metabolic oscillations. The novelty in this work, therefore, lies mainly in a description of dynamic metabolic behaviors in single bacterial cells.

I recommend publication if the comments below are all satisfactorily addressed.

Noteworthy results

E. coli cells exhibit glycolytic oscillations under specific growth conditions, like those previously described for the yeast *S. cerevisiae*.

The use of FRET-based metabolite sensors provides high temporal resolution of metabolic dynamics in single bacterial cells.

Pyruvate levels fluctuate in single isogenic cells grown in the same environment, highlighting the occurrence of non-genetic metabolic heterogeneity in bacteria.

NAD(P)H fluctuations show fluctuations of a similar frequency to pyruvate, suggesting that these oscillations are related.

In contrast to *S. cerevisiae*, metabolic oscillations appear not to involve PFK, but rather enzymes involved in Pyruvate metabolism (with a mechanistic interpretation, that unfortunately, is quite vague).

We thank the Reviewer for these overall positive comments on our work.

Major comments

- The core model is used to demonstrate the capacity of (*E. coli*) glycolysis to generate metabolic oscillations and to identify parameters that affect the model's propensity to produce these oscillations. However, we do not learn all that much about the mechanistic underpinnings and especially, how the regulatory (allosteric) architecture affects this behavior. Extending their current analysis to consider (generally) the sufficiency or necessity of specific feedbacks and the underlying architecture to generate the observed oscillations, would be informative (the structure of the network, incl allosteric interactions, may be more important than the choice of kinetic parameters).

This is a valid point, and we thank the Reviewer for raising it. To investigate whether oscillations depend on the model structure, in addition to the parameter analysis we

now systematically tested all different feedback mechanisms in our model (three allosteric feedback regulations result in eight possible combinations). Moreover, we added a stability criterium to the initial steady state that each model must satisfy. To switch off a feedback, we set the respective power-law term (a_1 , a_2 , a_3) to zero. For 10,000 simulated parameter sets, the models with at least two feedbacks, including the FBP feed forward activation of PYK, showed the propensity to produce oscillations.

-Some FRET-sensor controls appear to be lacking (and should be included for publication):

- o Control experiments with a non-responsive FRET-sensor are missing (e.g. pT162M115). This is needed to show that no other cellular parameters affect the sensor's response (Line 172, 184, 238).

We thank the Reviewer for pointing this out. We have performed the control experiments using the non-responsive FRET sensor (Supplementary Table 1). The *E. coli* cells with this mutant sensor did not show response (Supplementary Fig. 7a) or increased fluctuations (Supplementary Fig. 7b) when stimulated with 10 μ M glucose, indicating that the FRET responses of the pyruvate sensor and observed low-frequency fluctuations are indeed specific.

- o 2,4-DNP can significantly influence fluorescence ratios if not properly corrected for. How was the background contribution of 2,4-DNP accounted for? Here also, a non-responsive sensor would be needed.

Our results suggest that the observed change of fluorescence ratio for the pyruvate sensor in response to 2,4-DNP is indeed due to the FRET change, since the CFP signal decreased and YFP signal increased. We now comment on this in lines 204-209.

- o Did the authors calculate fluorescence bleed-through in their system, and was this corrected for? For reliable FRET-based measurements, this should be accounted for.

Although adjusting for the fluorescence bleed-through is indeed important when calculating the absolute levels of FRET, here we are only interested in the changes in FRET upon stimulation and not in its absolute levels. We could thus simply use the ratio between the CFP/YFP (or YFP/CFP) fluorescence as a proxy for FRET. We have previously shown that the change in this ratio is proportional to the number of interacting FRET pairs. For the monomolecular FRET sensor as used the same will hold for the number of sensors in an on-state. We apologize that this was not explained in the previous version of the manuscript and we thank the Reviewer for pointing this out; we now better justify why we use this FRET ratio in the Results and Methods (lines 167-169 and 742-746).

- Could the authors motivate their choice to sample from a (log)uniform distribution? Would a bell-shaped distribution (e.g. normal, log-normal) not be a better reflection of non-genetic metabolic heterogeneity? Given the large sampling range, are the parameters still biologically relevant, and if not could the authors motivate why this doesn't matter?

We used a (log)uniform distribution to sample the kinetic parameters, because this way small and large parameter values are equally represented. In a uniform distribution, in contrast, small values will be underrepresented, simply because there are more values between 1 and 10 than there are between 0.1 and 1. For example, to account for low and high enzyme saturation, we set the metabolite concentrations to one and sampled K_m values between 0.1 (high saturation) and 10 (low saturation). In this example, high enzyme saturation would be less frequent than low enzyme saturation.

We did not choose a normal distribution, because parameter values are a) mostly based on in-vitro studies that do not necessarily translate to in-vivo, and b) we use a simple model in which several reactions of the lower glycolysis are lumped together. However, we agree that in case of known parameter values, assuming a normal distribution around the true value would be the correct approach to take.

- Given how small the oscillation-inducing glucose range is ($< 100 \mu\text{M}$). How were concentrations for other carbon sources (Fig 4) chosen? It seems like these results may depend strongly on the exact concentration of the substrate. Without some variation in substrate concentration, the conclusion that some carbon sources induce oscillations and others don't is shaky. All that can be said, is that at the chosen concentrations these substrates don't induce oscillations (and they might at other concentrations). For example, if glucose experiments were done only at concentrations above 1 mM, the conclusion would be that glucose does not induce oscillations.

We apologize that we did not explain this clearly in the text of the original manuscript. The concentrations of the carbon sources and metabolites used for fluctuation measurements could elicit similar amplitudes of FRET responses to that of $10 \mu\text{M}$ glucose, which were below those of 1 mM-10 mM glucose that were saturated concentrations for glucose (Fig. 2g and Supplementary Figure 4), and lower concentrations did not elicit any changes in pyruvate levels. We now explained it in the revised manuscript (lines 193-197, 324-328 and 339-342).

Minor comments

- Showing population averaged only (e.g. fig S7C does not show whether or not single cells display fluctuation behaviors, I also miss here the autocorrelation and PSD plots). Furthermore, showing only 4 cells is prone to cherry-picking. Perhaps an appendix with

all the single-cell traces (at least for the most important figures, such as S7C, 3, 6C) or, more practically, a table indicating fractions of oscillating and non-oscillation single cells (also see next comment).

- Under conditions where oscillations are observed, it would be informative to indicate what fraction of single cell traces show oscillations and what fraction doesn't. I.e. is this a robust behaviour at the single cell level?

We thank the Reviewer for these suggestions. In our revision, we have shown the standard error of the mean of the individual PSDs on every plot (Figure 3e, Figure 4, Figure 5c,f,i, Supplementary Figure 5d, 7b, 8b, and 14d), as well as the distribution of the individual PSD values at a fixed low frequency for the most experiments (Supplementary Figure 6, 9, 11e,f, 14b,c, 15b,c, and 16b,c). This analysis confirmed the robustness of the fluctuation/oscillation behaviors at the single-cell level.

In the revised manuscript, we also included the autocorrelation curves for the fluctuations in different knockouts (Supplementary Figure 12a,b), and the fluctuations of PTS, cAMP, and NADH levels (Supplementary Figure 14a, 15a, and 16a).

- The authors mention that bleaching correction was applied for autocorrelation and PSD analyses. What about the other experiments? 1.0 s exposure seems long, and with repeated measurements, this can certainly lead to significant bleaching.

We only performed the bleaching correction for the PSD and autocorrelation analyses. We used 1.0 s exposure time together with ND4 and ND8 filters, so the bleaching for the measurements was very weak, as could be seen for example in Supplementary Figure 2b,d. Therefore, the traces shown for the CFP/YFP ratio have not been corrected for bleaching.

- Line 233: what is a majority? Please give a % or fraction.

We apologize for this inexact statement. Please see the response above. We have shown the standard error of the mean of the individual PSDs on every plot (Figure 3e) and the distribution of the individual PSD values at a fixed low frequency (Supplementary Figure 6).

- Line 378: can also be specifically induced by glucose? The use of a non-phosphorylatable EIIA_{glc} unit would be informative. Or measure the phosphorylation status in single cells? Or knockout and use another transport system?

We think that the fluctuations in responding to glucose using this PTS sensor should be specific, since this FRET pair has been already well characterized in our previous work at the population level, and also because pyruvate elevated the FRET signal (YFP/CFP

ratio) for this sensor but induced very weak fluctuations. We explained it now in lines 444-447 of the revised manuscript.

- Line 380-381: For experiments with the PTS FRET pair, the authors use TB medium for “consistency” with previous experiments. From line 379, M9-medium grown cells display much reduced fluctuations, which would imply that something in LB/TB is important to induce large fluctuations – could the authors comment on what this may be, and whether this has implications for the interpretation?

We are sorry that we did not describe the protocol clearly enough in the text. Similar to all other experiments, we did not grow cells in M9 medium, just equilibrated the LB/TB grown cells in M9 salts buffer and measured the FRET response and fluctuations in buffer with or without indicated carbon source. We explained it now better in lines 434-437.

- Line 391: do the authors see NADH fluctuations in the knockouts in S6g and S6h? And vice versa (the mutant in S6c?).

This is an interesting suggestion and we have now performed pilot experiments to measure changes in the NADH as well as its fluctuations upon exposure to glucose in $\Delta pykF$ and $\Delta ppc pykA pykF$ knockouts. However, we observed that the magnitude and the dynamics of NADH changes upon addition of glucose are very different between these strains and the wildtype. Because of these differences that we cannot easily explain, we would prefer not to include these experiments in the manuscript.

- Line 424: However, if such should read: However, whether such. In this sentence, “if” indicates a conditional, which is absent.

We have revise the text accordingly.

- Line 633: Cells with a FRET-ratio change of $< 10\%$ were considered unresponsive and discarded from analyses. Please provide an indication of the percentage/fraction of cells that were typically excluded. Does the inclusion/exclusion of these cells influence data interpretation? A small FRET-ratio does not necessarily indicate an unresponsive sensor, it could also just be that these cells do not experience large perturbations in pyruvate concentration.

The percentage of unresponsive cells was below 15%. We mention it now in the Methods section (lines 747-748). Exclusion of non-responsive cells did not influence the results.

- The link between the model and experiment could perhaps be made more explicit in the discussion. (it is only briefly discussed in the discussion, line 467). What is the model adding to the experiments, how well do experimental manipulations (e.g. knockouts) match model predictions?

We agree and thank the Reviewer for this suggestion. We have now substantially expanded the comparison between experiments and the model predictions in the revised text (lines 347-353 and lines 529-546).

- Could the authors comment on the observation that only PDH inactivation abolishes fluctuations? This step appears to exert a lot of control over the pathway behavior – what features are responsible for the dominant contribution of PDH (especially considering the conclusion that control appears to be distributed across several components).

We now comment on this in the Discussion section, as suggested (lines 547-553).

- It is sometimes unclear why the authors choose specific and varying substrate concentrations in different experiments. E.g. Fig 4 and Fig S7 *aceF* mutant. 1 mM glucose is used, as opposed to 10 μ M (in most other experiments). Especially, given the conclusion in Line 316-318 that deletion of *aceF* abolishes fluctuations (when exposed to 1 mM glucose), where experiments with WT showed that fluctuations are absent at millimolar (incl 1mM) concentrations. In these instances, it would be useful to state clearly the reasons for these differences in substrate concentration.

We apologize that it was not clearly explained in the previous version of the manuscript. Similar to our responses above regarding choices of concentrations of different substrates, we used 1 mM instead of 10 μ M glucose for the $\Delta aceF$ mutant because lower concentrations did not elicit FRET response in this strain. We explained it now in the revised manuscript (lines 386-388).

- Could the authors comment/speculate on the functional significance of differences in the oscillatory behavior seen for 10 μ M vs 1 and 10 mM glucose pulses (as shown in Fig 3)? (also see next comment)

As mentioned above in our responses to other referees, we fully agree that understanding the physiological significance of the observed fluctuations is highly interesting, and it will be the focus of our future work. What we can say at this point is that these fluctuations might affect not only NADH levels but also activity of the PTS, and therefore sugar uptake as well as the regulatory functions of the PTS. We now show additional experiments demonstrating that cAMP levels, one of the best-characterized

regulatory outputs of the PTS, also fluctuate. Since cAMP regulates gene expression, the latter might also be affected (although this was not tested yet).

- It is e.g. striking that 30 mM of fructose induces oscillations, whereas 1mM of glucose doesn't. What (kinetic/regulatory) differences in the underlying metabolic network explain this stark difference in sensitivity?

High concentrations of fructose were needed to induce measurable changes in the levels of pyruvate in cells grown in LB. We explained it in the text, lines 193-197 and 324-328. In contrast, for glucose 1 mM was far above saturation.

- In the abstract, the authors conclude that fluctuations in glycolysis “apparently propagate to other cellular processes, leading to ... heterogeneity of cellular states within the population”. The use of apparent implies observations that would support this statement. The data included here allows at best for speculation that fluctuations “could possibly propagate”. Not all fluctuations have a functional impact on phenotypic states.

We have revised the Abstract, but also added experiments on cAMP fluctuations that support this hypothesis.

REVIEWER COMMENTS

Reviewer #1 (Remarks to the Author):

The authors addressed all my comments. I also appreciated the clear responses to the other reviewer's comments and I have no further concerns.

Reviewer #2 (Remarks to the Author):

The authors addressed most of my technical concerns. In particular, I appreciate the new control experiments, clarification and analysis on structural perturbations of the model. However, in my view the main limitations of this study being an observative study remain. The model doesn't add much to the main observation of pyruvate oscillations (as also stated by the authors even large structural perturbations of the model would still yield pervasive oscillations) and I'm still wondering what is that we actually learn.

Reviewer #3 (Remarks to the Author):

The authors have made a sincere effort to address the concerns of the reviewers, including additional analyses and experiments, with adjustments and additions both clarifying and strengthening the findings presented. The most important additions, include (1) a systematic computational analysis of the necessity and/or essentiality of three primary allosteric feedbacks to generate oscillations in pyruvate concentrations, (2) the inclusion of a non-responsive Pyruvate FRET-sensor and (3) the quantification of cAMP fluctuations. While the first two additions, strengthen the core claims of the manuscript, the last provides a link to essential regulatory signaling processes, and thereby a potential functional explanation for the observed oscillations (although the current data is insufficient to go beyond speculation of a functional link).

There are, however, still three points that I would like to raise, with only the first being significant.

The first concerns a control experiment (pointed to in the original review) to account for the effects of 2,4-DNP on FRET signals. Although the authors address this point in their rebuttal, their response (and the data they offer) does not sufficiently address this concern. They point to supplement Fig 2d, indicating that CFP signal decreases while YFP increases, noting that this indicates a FRET-specific response. Unfortunately, the changes in CFP and YFP are insufficient to exclude an effect of DNP. DNP

can still have a differential effect on the CFP and YFP signals, respectively. In our own FRET-based metabolite sensor experiments, DNP causes a significant decrease in the FRET ratio of a non-responsive sensor (in our case for a CFP/RFP pair). What we see is a decrease in signal in both the CFP and RFP channels, but a much larger decrease in the CFP signal (leading to an overall decreased CFP/RFP ratio); this initially led (before we corrected for this effect) to incorrect conclusions regarding the magnitude of perturbations that cause changes in the FRET-ratio. Given that DNP is yellowish in color, it cannot be excluded that the observed change in FRET ratio isn't (partly) caused by DNP, due to a decrease in CFP signal (which we know DNP causes) and an increase in YFP (which admittedly we haven't specially tested, but cannot be dismissed). To exclude a possible DNP-related artifact, I would strongly suggest a very simple control experiment, using the same non-responsive sensor the authors now included in the revised manuscript. A plot showing the individual CFP and YFP time traces and the corresponding FRET ratio trace can be included in the supplement for comparison with the responsive sensor. While I don't think the core conclusion of the manuscript will stand or fall on this experiment, I do feel that this is a simple control that should be done.

The second and third points (both minor) concern two rather striking observations, which are only briefly (and or generally) considered in the text. These points are not significant, and as such, not essential to address (but perhaps something to consider in follow-up projects).

- The authors now comment on the impact of PDH inactivation on oscillations in the discussion(547-553). It is a pity that no further speculative consideration regarding the possible reasons for this profound impact ("Inactivation of PDH was the only identified mutation that completely abolished pyruvate fluctuations in our experiments") is offered.

- The authors make clear, in the revised manuscripts, that different substrates can show large differences in the concentrations at which they impact a common metabolic component/process. While I fully appreciate this point, the fructose result is difficult to understand. Is the metabolism of fructose and glucose (including uptake kinetics and growth rate) really so different that it could explain the ~300-fold difference in the concentration required to induce oscillations (100 μ M vs 30 mM)? It seems that the striking difference, in this specific instance (glucose vs fructose), should receive somewhat more consideration than simply putting it down to differences in uptake in LB medium (which I strongly doubt explains the fructose case).

Point-by-point response to reviewers' comments

We would like to thank the reviewers again for their detailed feedback on our manuscript and suggestions for its improvement. We are very glad to see that the reviewers are largely satisfied with our response to their previous comments and with our changes made in the revised manuscript. Below are our detailed responses to the remaining points of individual reviewers:

Reviewer #1 (Remarks to the Author):

The authors addressed all my comments. I also appreciated the clear responses to the other reviewer's comments and I have no further concerns.

We thank the Reviewer for this very positive feedback, and we are happy to see that we could satisfactorily address all comments of this Reviewer.

Reviewer #2 (Remarks to the Author):

The authors addressed most of my technical concerns. In particular, I appreciate the new control experiments, clarification and analysis on structural perturbations of the model. However, in my view the main limitations of this study being an observative study remain. The model doesn't add much to the main observation of pyruvate oscillations (as also stated by the authors even large structural perturbations of the model would still yield pervasive oscillations) and I'm still wondering what is that we actually learn.

We thank the Reviewer for appreciating the additional experimental and modeling work added to the revised manuscript and we are glad that it could address technical concerns of this Reviewer. Regarding the limitations of our study mentioned by the Reviewer: Although identifying the exact molecular mechanism behind the observed oscillations has indeed proven to be difficult, both our model and experiments suggest that it is because of the complex multifactorial nature of this phenomenon. Nevertheless, we systematically identified the reactions that contribute to and influence these dynamics (and those that do not), thus clearly going beyond a purely observative study. But given that our work presents the first clear description of metabolic oscillations in bacteria, there is obviously still much to be learned about this phenomenon, both about its mechanisms and about its physiological importance, which cannot all be covered in a single paper.

Reviewer #3 (Remarks to the Author):

The authors have made a sincere effort to address the concerns of the reviewers, including additional analyses and experiments, with adjustments and additions both clarifying and strengthening the findings presented. The most important additions, include (1) a systematic computational analysis of the necessity and/or essentiality of three primary allosteric feedbacks to generate oscillations in pyruvate concentrations, (2) the inclusion of a non-responsive Pyruvate FRET-sensor and (3) the quantification of cAMP fluctuations. While the first two additions, strengthen the core claims of the manuscript, the last provides a link to essential regulatory signaling processes, and thereby a potential functional explanation for the observed oscillations (although the current data is insufficient to go beyond speculation of a functional link).

There are, however, still three points that I would like to raise, with only the first being significant.

We thank the Reviewer for this very positive feedback and for acknowledging that the additions made to the revised manuscript have further strengthened and clarified our findings. We agree with the Reviewer that the functional implications of the observed oscillations for bacterial cell physiology require further investigation, which will be the aim of our follow-up work. We have now addressed remaining points raised by the reviewer:

The first concerns a control experiment (pointed to in the original review) to account for the effects of 2,4-DNP on FRET signals. Although the authors address this point in their rebuttal, their response (and the data they offer) does not sufficiently address this concern. They point to supplement Fig 2d, indicating that CFP signal decreases while YFP increases, noting that this indicates a FRET-specific response. Unfortunately, the changes in CFP and YFP are insufficient to exclude an effect of DNP. DNP can still have a differential effect on the CFP and YFP signals, respectively. In our own FRET-based metabolite sensor experiments, DNP causes a significant decrease in the FRET ratio of a non-responsive sensor (in our case for a CFP/RFP pair). What we see is a decrease in signal in both the CFP and RFP channels, but a much larger decrease in the CFP signal (leading to an overall decreased CFP/RFP ratio); this initially led (before we corrected for this effect) to incorrect conclusions regarding the magnitude of perturbations that cause changes in the FRET-ratio. Given that DNP is yellowish in color, it cannot be excluded that the observed change in FRET ratio isn't (partly) caused by DNP, due to a decrease in CFP signal (which we know DNP causes) and an increase in YFP (which admittedly we haven't specially tested, but cannot be dismissed). To exclude a possible DNP-related artifact, I would strongly suggest a very simple control experiment, using the same non-responsive sensor the authors now included in the revised manuscript. A plot showing the individual CFP and YFP time traces and the corresponding FRET ratio trace can be included in the supplement for comparison with the responsive sensor. While I don't think the core conclusion of the manuscript will

stand or fall on this experiment, I do feel that this is a simple control that should be done.

We thank the Reviewer for raising this point. We have now performed additional control experiments with DNP, as suggested by the Reviewer, and indeed observed that DNP directly affects fluorescence. At least in our microscopy system, we see that for both tested negative controls, either the FRET sensor with strongly reduced sensitivity to pyruvate or a direct fusion between CFP and YFP, the addition of DNP increases fluorescence in both channels. These data for individual channels and the ratio in control experiments are now shown in Supplementary Figure 2e,f as suggested. This unspecific increase in fluorescence of both channels indeed precludes exact quantification of true changes in the FRET signal upon exposure to DNP (as suggested by this Reviewer), particularly since it also affects the ratio of CFP to YFP fluorescence. Nevertheless, the response of the functional sensor was different, with opposing changes in the CFP and YFP channel fluorescence, which was true not only for the DNP measurement shown in Supplementary Figure 2d but also for other measurements where DNP was used, shown in Supplementary Figure 11d and Supplementary Figure 12d. This difference from the negative controls suggests that the functional sensor does detect a decrease in pyruvate upon the DNP treatment, on top of the unspecific effect of DNP. Thus, we can still draw a qualitative conclusion about the ability of this sensor to respond to a downregulation of pyruvate levels, which was the only purpose of this experiment with DNP (and as mentioned by this Reviewer, this statement is anyway not critical for drawing other conclusions in our manuscript). We have now revised the text of the manuscript accordingly (lines 216-239 in the version of the manuscript with changes tracked; lines 206-219 in the clean version of the revised manuscript), to mention the uncertainties in the quantification of the FRET response to DNP due to its unspecific effects on fluorescence, and to highlight that we can only draw a qualitative conclusion based on this response.

The second and third points (both minor) concern two rather striking observations, which are only briefly (and or generally) considered in the text. These points are not significant, and as such, not essential to address (but perhaps something to consider in follow-up projects).

- The authors now comment on the impact of PDH inactivation on oscillations in the discussion(547-553). It is a pity that no further speculative consideration regarding the possible reasons for this profound impact ("Inactivation of PDH was the only identified mutation that completely abolished pyruvate fluctuations in our experiments") is offered.

We thank the Reviewer for this suggestion to expand the discussion about the loss of oscillations upon inactivation of PDH. Deletion of *aceF* (PDH component) is expected to increase the steady-state level of intracellular pyruvate, which is indeed supported by our observation of higher FRET ratio in buffer-equilibrated cells (Supplementary Fig. 12d). Such high level of intracellular pyruvate might either suppress or mask the

pyruvate fluctuations. We now mention it in the Results (lines 428-430 in the version of the manuscript with changes tracked; lines 390-393 in the clean version of the revised manuscript) and in the Discussion (lines 604-609 in the version of the manuscript with changes tracked; lines 560-564 in the clean version of the revised manuscript).

- The authors make clear, in the revised manuscripts, that different substrates can show large differences in the concentrations at which they impact a common metabolic component/process. While I fully appreciate this point, the fructose result is difficult to understand. Is the metabolism of fructose and glucose (including uptake kinetics and growth rate) really so different that it could explain the ~300-fold difference in the concentration required to induce oscillations (100 μ M vs 30 mM)? It seems that the striking difference, in this specific instance (glucose vs fructose), should receive somewhat more consideration than simply putting it down to differences in uptake in LB medium (which I strongly doubt explains the fructose case).

We apologize for not elaborating more clearly on this difference. As mentioned, only high concentration of fructose, in the millimolar range, triggered elevation in the level of pyruvate that was measurable by FRET (and induced pyruvate oscillations) under our conditions. Although further experiments would be required to elucidate the reason for this concentration difference between stimulation with fructose and glucose, it might be either due to poor expression of the fructose uptake system in cells grown in LB or because of a different entry point of fructose in glycolysis (via FBP). The same reasoning may be applicable to other alternative metabolites used in our study, which all enter glycolysis at lower points. We now comment on these possible explanations, and mention that we do not have a definitive answer, when describing stimulation of FRET response with different metabolites (lines 201-207 in the version of the manuscript with changes tracked; lines 192-198 in the clean version of the revised manuscript) and also when describing induction of pyruvate fluctuations (lines 368-370 in the version of the manuscript with changes tracked; lines 335-340 in the clean version of the revised manuscript).